# Counting Substructures with Higher-Order Graph Neural Networks: Possibility and Impossibility Results

## Abstract

While message passing Graph Neural Networks (GNNs) have become increasingly popular architectures for learning with graphs, recent works have revealed important shortcomings in their expressive power. In response, several higher-order GNNs have been proposed that substantially increase the expressive power, albeit at a large computational cost. Motivated by this gap, we explore alternative strategies and lower bounds. In particular, we analyze a new recursive pooling technique of local neighborhoods that allows different tradeoffs of computational cost and expressive power. First, we prove that this model can count subgraphs of size $k$, and thereby overcomes a known limitation of low-order GNNs. Second, we show how recursive pooling can exploit sparsity to reduce the computational complexity compared to the existing higher-order GNNs. More generally, we provide a (near) matching information-theoretic lower bound for counting subgraphs with graph representations that pool over representations of derived (sub-)graphs. We also discuss lower bounds on time complexity.

## 1 Introduction

Graph Neural Networks (GNNs) are powerful tools for graph representation learning (Scarselli et al., 2008; Kipf & Welling, 2017; Hamilton et al., 2017), and have been successfully applied to molecule property prediction, simulating physics, social network analysis, knowledge graphs, traffic prediction and many other domains (Duvenaud et al., 2015; Defferrard et al., 2016; Battaglia et al., 2016; Jin et al., 2018). The perhaps most widely used class of GNNs, Message Passing Graph Neural Networks (MPNNs) (Gilmer et al., 2017; Kipf & Welling, 2017; Hamilton et al., 2017; Xu et al., 2019; Scarselli et al., 2008), follow an iterative message passing scheme to compute a graph representation.

Despite the empirical success of MPNNs, their expressive power has been shown to be limited. For example, their discriminative power, at best, corresponds to the one-dimensional Weisfeiler-Leman (1-WL) graph isomorphism test (Xu et al., 2019; Morris et al., 2019), so they cannot distinguish regular graphs, for instance. Likewise, they cannot count any induced subgraph with at least three vertices (Chen et al., 2020), or learn structural graph parameters such as clique information, diameter, conjoint or shortest cycle (Garg et al., 2020). Yet, in applications like computational chemistry, materials design or pharmacy (Elton et al., 2019; Sun et al., 2020; Jin et al., 2018), the functions we aim to learn often depend on the presence or count of specific substructures, e.g., functional groups.

The limitations of MPNNs result from their inability to distinguish individual nodes. To resolve this issue, two routes have been studied: (1) using unique node identifiers (Loukas, 2019; Sato et al., 2019; Abboud et al., 2021), and (2) higher-order GNNs that act on $k$-tuples of nodes. Node IDs, if available, enable Turing completeness for sufficiently large MPNNs (Loukas, 2019). Higher-order networks use an encoding of $k$-tuples and then apply message passing (Morris et al., 2019), or equivariant tensor operations (Maron et al., 2018).

The expressive power of MPNNs is often measured in terms of a hierarchy of graph isomorphism tests, specifically the $k$-Weisfeiler-Leman ($k$-WL) hierarchy. The $k$-order models in (Maron et al., 2018) and (Maron et al., 2019a) are equivalent to the $k$-WL and $(k+1)$-WL "tests", respectively, and are universal for the corresponding function classes (Azizian & Lelarge, 2020; Maron et al., 2019b; Keriven & Peyré, 2019). Yet, these models are computationally expensive, operating on $\Theta(n^k)$

tuples and according to current upper bounds requiring up to $O(n^k)$ iterations (Kiefer & McKay, 2020). The necessary tradeoffs between expressive power and computational complexity are still an open question. However, for specific classes of tasks this full universality may not be needed. Here, we study such an example of practical interest: counting substructures, as proposed in (Chen et al., 2020). In particular, we study if it is possible to count given substructures with a GNN whose complexity is between that of MPNNs and existing higher-order GNNs.

To this end, we study a generic scheme followed by many GNN architectures, including MPNNs and higher-order GNNs (Morris et al., 2019; Chen et al., 2020): select a collection of subgraphs of the input graph, encode these, and apply an aggregation function on this collection. First, we study the power of pooling *by itself*, as a multi-set function over node features. We prove that $k$ *recursive* applications on each node's neighborhood allow to count any substructure of size $k$. This is in contrast to *iterative* MPNNs. We call this technique *Recursive neighborhood pooling (RNP)*. While subgraph pooling relates to the graph reconstruction conjecture, our strategy has important differences. In particular, we show how the aggregation "augments" local encodings, if they play together and the subgraphs are selected appropriately, and this reasoning may be of interest for the design of other, even partially, expressive architectures. Moreover, our results show that the complexity is *adjustable* to the counting task of interest and the sparsity of the graph.

The strategy of pooling subgraph encodings has previously been used for counting in Local Relational Pooling (LRP) (Chen et al., 2020). LRP relies on an isomorphic encoding of subgraphs, which is expensive – e.g., the relational pooling it uses requires $O(k!)$ time for a subgraph of size $k$. Other higher-order GNNs would be expensive, too, as high orders are needed for complete isomorphism power. A major difference to our RNP is that our recursion uses subgraphs of *varying* sizes and structures, many of them much smaller – adapted to the graph structure and specific counting task.

Furthermore, we study lower bounds on GNNs that count motifs. We show an information-theoretic lower bound on the number of subgraphs to encode, as a function of an encoding complexity. We also transfer computational lower bounds that apply to any counting GNN. The lower bounds show that the recursive pooling is close to tight.

In short, in this paper, we make the following contributions:

- We study the power of pooling encodings of subgraphs, and show that pooling, as an injective multi-set function, is sufficient *by itself* for counting when applied *recursively* on appropriate subgraphs, remarkably without relying on other encoding techniques or node IDs. This is different from any other strategy we are aware of in the literature.
- We analyze the complexity of recursive pooling, as a function of the task and input graph.
- We provide complexity lower bounds for pooling and general GNN architectures that count motifs. For instance, we show a lower bound on the number of subgraphs that need to be encoded.

## 2 BACKGROUND

**Message Passing Graph Neural Networks.** Let $G = (\mathcal{V}, \mathcal{E}, X)$ be an attributed graph with $|\mathcal{V}| = n$ nodes. Here, $X_v \in \mathcal{X}$ denotes the initial attribute of $v \in \mathcal{V}$, where $\mathcal{X} \subseteq \mathbb{N}$ is a (countable) domain. A typical Message Passing Graph Neural Network (MPNN) first computes a representation of each node, and then aggregates the node representations via a readout function into a representation of the entire graph $G$. The representation $h_v^{(i)}$ of each node $v \in \mathcal{V}$ is computed iteratively by aggregating the representations $h_u^{(i-1)}$ of the neighboring vertices $u$:

$$m_v^{(i)} = \text{AGGREGATE}^{(i)}\left(\{\!\!\{h_u^{(i-1)} : u \in \mathcal{N}(v)\}\!\!\}\right), \quad h_v^{(i)} = \text{COMBINE}^{(i)}\left(h_v^{(i-1)}, m_v^{(i)}\right), \quad (1)$$

for any $v \in \mathcal{V}$, for $k$ iterations, and with $h_v^{(0)} = X_v$. The AGGREGATE/COMBINE functions are parametrized, and $\{\!\!\{.\}\!\!\}$ denotes a multi-set, i.e., a set with (possibly) repeating elements. A graph-level representation can be computed as $h_G = \text{READOUT}\left(\{\!\!\{h_v^{(k)} : v \in \mathcal{V}\}\!\!\}\right)$, where READOUT is a learnable aggregation function. For representational power, it is important that the learnable functions are injective (Xu et al., 2019).

**Higher-Order GNNs.** To increase the representational power of GNNs, several higher-order GNNs have been proposed. In $k$-*GNN*, message passing is applied to $k-$tuples of nodes, inspired

by $k$-WL (Morris et al., 2019). At initialization, each $k$-tuple is labeled such that two $k$-tuples are labeled differently if their induced subgraphs have different isomorphism types (Maron et al., 2019a; Cai et al., 1992). As a result, $k$-GNNs can count (induced) substructures with at most $k$ vertices even at initialization. Another class of higher-order networks applies (linear) equivariant operations, interleaved with coordinate-wise nonlinearities, to order-$k$ tensors consisting of the adjacency matrix and input node attributes (Maron et al., 2018; 2019a;b). These GNNs are at least as powerful as $k-$GNNs, and hence they too can count substructures with at most $k$ vertices. All these methods need $\Omega(n^k)$ operations. *Local Relational Pooling (LRP)* (Chen et al., 2020) was designed for counting and applies relational pooling (Murphy et al., 2019b;a) on local neighborhoods, i.e., one pools over evaluations of a permutation-sensitive function applied to all $k!$ permutations of the nodes in a $k$-size neighborhood of each node.

## 3 OTHER RELATED WORKS

**Expressive power.** Several other works have studied the expressive power of GNNs as function approximators (Azizian & Lelarge, 2020). Scarselli et al. (2009) extend universal approximation from feedforward networks to MPNNs, using the notion of *unfolding equivalence*, i.e., functions on computation trees. Indeed, graph distinction and function approximation are closely related (Chen et al., 2019; Azizian & Lelarge, 2020; Keriven & Peyré, 2019). Maron et al. (2019b) and Keriven & Peyré (2019) show that higher-order, tensor-based GNNs provably achieve universal approximation of permutation-invariant functions on graphs, and Loukas (2019) analyzes expressive power under depth and width restrictions. Studying GNNs from the perspective of local algorithms, Sato et al. (2019) show that GNNs can approximate solutions to certain combinatorial optimization problems. For counting substricutres, Arvind et al. (2020); Fürer (2017) show that 2-WL, which is equivalent to MPNNs, can count not necessarily induced cycles and paths of up to 7 vertices with $O(n^2)$ complexity,

**Subgraphs and GNNs.** Having infromation about subgraphs can be quite helpful in various graph representation algorithms (Liu et al., 2019; Monti et al., 2018; Liu et al., 2020; Yu et al., 2020; Meng et al., 2018; Cotta et al., 2020; Alsentzer et al., 2020; Huang & Zitnik, 2020). For example, for graph comparison (i.e., testing whether a given (possibly large) subgraph exists in the given model), Ying et al. (2020) compare the outputs of GNNs for small subgraphs of the two graphs. To improve the expressive power of GNNs, Bouritsas et al. (2020) use features that are counts of specific subgraphs of interest. Another example is (Vignac et al., 2020), where an MPNN is strengthened by learning local context matrices around vertices. Recent works have also developed GNNs that pass messages on ego-nets (You et al., 2021; Sandfelder et al., 2021). With motivation from the reconstruction conjecture, Cotta et al. (2021) process node-deleted subgraphs with individual MPNNs, and then pool them with a DeepSets model to get a representation of the original graph.

## 4 RECURSIVE NEIGHBORHOOD POOLING

Let $G = (\mathcal{V}, \mathcal{E}, X)$ be an attributed input graph with $|\mathcal{V}| = n$ nodes, and let $h_v^{(0)} = X_v$ be the initial representation of each node $v$. In this work, we study architectures that first find representations of a collection of $m$ subgraphs $G_i$ and then aggregate (pool) over these representations with a multi-set function, i.e.,

$$\text{AGGREGATE}(\{\!\!\{\psi(G_i) : i \in [m]\}\!\!\}), \quad [m] = \{1, \dots, m\}. \tag{2}$$

It is clear that if the $\psi$ can count a subgraph structure $H$, then the entire model can count $H$. In particular, we aim to apply this strategy to obtain the representations $\psi$, too. To appreciate the challenges in doing so, recall two such examples. First, MPNNs follow this pooling strategy, by iteratively aggregating over node neighborhoods, and then aggregating all node representations into a graph representation. However, it is known that MPNNs can count at most star structures or edges (Arvind et al., 2020). This is because they represent a local computation tree, which loses structural information about node identities and connectivity. Second, this strategy is at the heart of the Graph Reconstruction Conjecture (Kelly, 1957), which conjectures that a graph $G$ can be reconstructed from its subgraphs $\{\!\!\{G_v = G \setminus \{v\} : v \in V(G)\}\!\!\}$ (Appendix E). This, however, is unknown for general graphs. Although the $G_v$ retain some structure, we lose information about their *structural relationship*. In summary, encoding structural information is the key question.

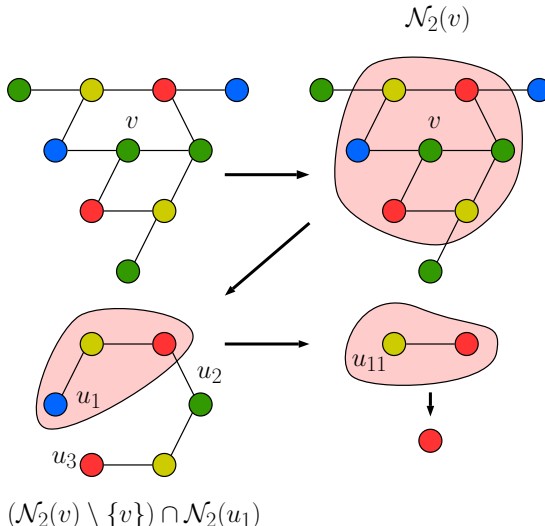

$\mathcal{N}_2(v)$

$(\mathcal{N}_2(v) \setminus \{v\}) \cap \mathcal{N}_2(u_1)$

Figure 1: Illustration of a Recursive Neighborhood Pooling GNN (RNP-GNN) with recursion parameters $(2, 2, 1)$. To compute the representation of node $v$ in the given input graph (depicted in the top left of the figure), we first recurse on $G(\mathcal{N}_2(v) \setminus \{v\})$ (top right of figure). To do so, we find the representation of each node $u \in G(\mathcal{N}_2(v) \setminus \{v\})$. For instance, to compute the representation of $u_1$, we apply an RNP-GNN with recursion parameters $(2, 1)$ and aggregate $G((\mathcal{N}_2(v) \setminus \{v\}) \cap (\mathcal{N}_2(u_1) \setminus \{u_1\}))$, which is shown in the bottom left of the figure. To do so, we recursively apply an RNP-GNN with recursion parameter $(1)$ on $G((\mathcal{N}_2(v) \setminus \{v\}) \cap (\mathcal{N}_2(u_1) \setminus \{u_1\}) \cap (\mathcal{N}_1(u_{11}) \setminus \{u_{11}\}))$, in the bottom right of the figure.

Hence, to represent the counting function $\psi$ over potentially large neighborhoods, an MPNN does not suffice. But, aggregation over input node attributes, along with edge information, can count edge types, i.e., tiny subgraphs. Hence, we *recursively* apply aggregation on smaller sub-neighborhoods while remembering structural information, with node-wise aggregation as the base case. For intuition, suppose we aim to count the occurrence of subgraph $H$ in the $r_1$-neighborhood $\mathcal{N}_{r_1}(v)$ of a node $v$. To do so, we may count $H_v = H \setminus \{v\}$ in the smaller graph $\mathcal{N}_{r_1}(v) \setminus \{v\}$. But, to combine these counts with the presence of $v$ to "complete" $H$, we must know how the $H_v$ are connected to $v$ in the screened graph. Hence, to retain structure information, we mark the neighbors of $v$. We then recursively call neighborhood pooling to process smaller neighborhoods $\mathcal{N}_{r_2}(u)$ *within* $\mathcal{N}_{r_1}(v) \setminus \{v\}$. This could, e.g., learn to count marked versions of $H_v$. The radii $r$ of neighborhoods may differ in recursive calls. In Section 5, we relate their size to $H$.

*Recursive neighborhood pooling* RNP-GNN$(G, \{h_u^{\text{in}}\}_{u \in \mathcal{V}(G)}, (r_1, \dots, r_\tau))$ takes an attributed graph along with a sequence of neighborhood radii for different recursions, and returns a set of node encodings $\{h_v\}_{v \in \mathcal{V}(G)}$. For any $v \in G$, RNP-GNN first constructs $v$'s neighborhood, removes $v$ and marks its neighbors[1]:

$$G_v \leftarrow \mathcal{N}_{r_1}(v) \setminus \{v\}, \qquad h_{u,\text{aug}}^{\text{in}} = (h_u^{\text{in}}, \mathbb{1}[(u, v) \in \mathcal{E}(G_v)]). \tag{3}$$

Here, we extracted $r_1-$neighborhood of $v$, removed the central node $v$, and then added the new structural information to $1-$neighbors of $v$. Then we aggregate over subgraph representations, i.e., the representations of $G_v, v \in [n]$ when they considered as a set graphs. If $\tau = 1$ (base case), we use the input features:

$$h_v \leftarrow \text{AGGREGATE}^{(\tau)}(h_v^{\text{in}}, \{\!\!\{h_{u,\text{aug}}^{\text{in}} : u \in G_v\}\!\!\}). \tag{4}$$

In other words, in this case we only combine hidden representations in each $G_v$ without any iteration/recursion. If $\tau > 1$, we recursively represent neighborhoods of nodes in $G_v$:

$$\{\hat{h}_{v,u}\}_{u \in G'} \leftarrow \text{RNP-GNN}\big(G_v, \{h_{u,\text{aug}}^{\text{in}}\}_{u \in G_v}, (r_2, r_3, \dots, r_\tau)\big) \tag{5}$$

$$h_v \leftarrow \text{AGGREGATE}^{(\tau)}\Big(h_v^{\text{in}}, \{\!\!\{\hat{h}_{u,v} : u \in G_v\}\!\!\}\Big). \tag{6}$$

Here, we first apply an RNP-GNN on each $G_v$ to update the representations of nodes for each node in $G_v$. Then, we combine those representations to find the representation of $v$. The fact that $G_v$ has less nodes than $G$ allows to define the algorithm in an inductive way; this ensures the algorithm does not have any logical loop. For aggregation, we can use, e.g., the injective multi-set function from (Xu et al., 2019):

$$\text{AGGREGATE}^{(\tau)}\Big(h_v, \{h_u\}_{u \in G_v}\Big) = \text{MLP}^{(\tau)}\Big((1 + \epsilon)h_v + \sum\nolimits_{u \in G_v} \hat{h}_u\Big), \tag{7}$$

---

[1] $\mathbb{1}[.]$ is one when the condition is satisfied, and otherwise is zero.

where we use MLP modules with ReLU activation, and $\epsilon$ is an arbitrary irrational constant (Xu et al., 2019). The final readout aggregates over the final node representations of the entire graph. Figure 1 illustrates an RNP-GNN with recursion parameters $(2, 2, 1)$, and Appendix F provides pseudocode. One can optionally do message passing iterations to the representations $h_v, v \in [n]$ to make the model more expressive; though our results hold even without those iterations.

While MPNNs also encode a representation of a local neighborhood, the recursive representations differ as they take into account *intersections* of neighborhoods. As a result, as we will see in Section 5, they retain more structural information and are more expressive than MPNNs; one can take $r_1 = 1$ to simulate one iteration of MPNNs (and then iterate).

Models like $k$-GNN and LRP also compute encodings of subgraphs, and then update the resulting representations via message passing. We can do the same with the neighborhood representations computed by RNP-GNNs to encode more global information, although our representation results in Section 5 hold even without that.

## 5    EXPRESSIVE POWER OF RECURSIVE POOLING

In this section, we analyze the expressive power of RNP-GNNs. We notice that for full expressiveness, we indeed need the identifiers in Eq. (3); their addition is an important insight. Still, RNP-GNN does maintain some expressive power without the augmented identifiers. For instance, consider the graphs $G_1$ = two triangles and $G_2$ = the 6-cycle, which are 1-WL equivalent and thus cannot be distinguished by MPNNs. An RNP-GNN with $r_1 = 1$ can distinguish these without the augmented identifiers. This is because on the first recursion level, the 1-neighborhoods of $G_1$ are two nodes with an edge between them, and the 1-neighborhoods of $G_2$ consist of two isolated nodes.

### 5.1    COUNTING (INDUCED) SUBSTRUCTURES

In contrast to MPNNs, which in general cannot count substructures of three vertices or more (Chen et al., 2020), in this section we prove that for any set of substructures, there is an RNP-GNN that provably counts them. We begin with a few definitions.

**Definition 1.** *Let $G, H$ be arbitrary, potentially attributed simple graphs, where $\mathcal{V}$ is the set of nodes in $G$. Also, for any $\mathcal{S} \subseteq \mathcal{V}$, let $G(\mathcal{S})$ denote the subgraph of $G$ induced by $\mathcal{S}$. The* induced subgraph count function *is defined as*

$$C(G; H) := \sum\nolimits_{\mathcal{S} \subseteq \mathcal{V}} \mathbb{1}\{G(\mathcal{S}) \cong H\}, \tag{8}$$

*i.e., the number of subgraphs of $G$ isomorphic to $H$.*

To relate the size of encoded neighborhoods to the substructure $H$, we will need a notion of *covering sequences* for graphs.

**Definition 2.** *Let $H = (\mathcal{V}_H, \mathcal{E}_H)$ be a simple connected graph. For any $\mathcal{S} \subseteq \mathcal{V}_H$ and $v \in \mathcal{V}_H$, define the covering distance of $v$ from $\mathcal{S}$ as*

$$\bar{d}_H(v; \mathcal{S}) := \max_{u \in \mathcal{S}} d(u, v), \tag{9}$$

*where $d(.,.)$ is the shortest-path distance in $H$.*

**Definition 3.** *Let $H$ be a simple connected graph on $\tau + 1$ vertices. A permutation of vertices, such as $(v_1, v_2, \ldots, v_{\tau+1})$, is called a* vertex covering sequence *with respect to a sequence* $\mathbf{r} = (r_1, r_2, \ldots, r_\tau) \in \mathbb{N}^\tau$ *called a* covering sequence, *if and only if*

$$\bar{d}_{H_i'}(v_i; \mathcal{S}_i) \leq r_i, \tag{10}$$

*for any $i \in [\tau + 1] = \{1, 2, \ldots, \tau + 1\}$, where $\mathcal{S}_i = \{v_i, v_{i+1} \ldots, v_{\tau+1}\}$ and $H_i' = H(\mathcal{S}_i)$ is the subgraph of $H$ induced by the set of vertices $\mathcal{S}_i$. We also say that $H$ admits* the covering sequence $\mathbf{r} = (r_1, r_2, \ldots, r_\tau) \in \mathbb{N}^\tau$ *if there is a vertex covering sequence for $H$ with respect to $\mathbf{r}$.*

In particular, in a covering sequence we first consider the whole graph as a local neighborhood of one of its nodes with radius $r_1$. Then, we remove that node and compute the covering sequence of

the remaining graph. Figure 4 shows an example of covering sequence computation. An important property, which holds by definition, is that if $\mathbf{r}$ is a covering sequence for $H$, then any $\mathbf{r}' \geq \mathbf{r}$ (coordinate-wise) is also a covering sequence for $H$.

Note that any connected graph on $k$ nodes admits at least one covering sequence, which is $(k-1, k-2, \ldots, 1)$. To observe this fact, note that in a connected graph, there is at least one node that can be removed and the remaining graph still remains connected. Therefore, we may take this node as the first element of a vertex covering sequence, and inductively find the other elements. Since the diameter of a connected graph with $k$ vertices is always bounded by $k-1$, we achieve the desired result. However, we will see in the next section that, when using covering sequences to identify sufficiently powerful RNP-GNNs, it is desirable to have covering sequences with low $r_1$, since the complexity of the resulting RNP-GNN depends on $r_1$.

More generally, if $H_1$ and $H_2$ are (possibly attributed) simple graphs on $k$ nodes and $H_1 \Subset H_2$, i.e., $H_1$ is a subgraph of $H_2$ (not necessarily induced subgraph), then it follows from the definition that any covering sequence for $H_1$ is also a covering sequence for $H_2$. As a side remark, as illustrated in Figure 2, covering sequences need not always to be decreasing. Using covering sequences, we can show the following result.

**Theorem 1.** *Consider a set of (possibly attributed) graphs $\mathcal{H}$ on $\tau+1$ vertices, such that any $H \in \mathcal{H}$ admits the covering sequence $(r_1, r_2, \ldots, r_\tau)$. Then, there is an RNP-GNN $f(\cdot; \theta)$ with recursion parameters $(r_1, r_2, \ldots, r_\tau)$ that can count any $H \in \mathcal{H}$. In other words, for any two graphs $G_1, G_2$, if there exists $H \in \mathcal{H}$ such that $C(G_1; H) \neq C(G_2; H)$, then $f(G_1; \theta) \neq f(G_2; \theta)$. The same result also holds for the non-induced subgraph count function.*

Theorem 1 states that, with appropriate recursion parameters, any set of (possibly attributed) substructures can be counted by an RNP-GNN. Interestingly, induced and non-induced subgraphs can be both counted in RNP-GNNs[2]. Also, for a given covering sequence, we can *simultaneously* count a set of substructures that each admit it. This means that one RNP-GNN is able to potentially count many substructures. We prove Theorem 1 in Appendix A.2. The main idea is to show that we can implement the intuition for recursive pooling outlined in Section 4 formally with the proposed architecture and multiset functions[3].

The theorem holds for any covering sequence that is valid for all graphs in $\mathcal{H}$. For any graph, one can compute a covering sequence by computing a spanning tree, and sequentially pruning the leaves of the tree. The resulting sequence of nodes is a vertex covering sequence, and the corresponding covering sequence can be obtained from the tree too (Appendix D). A valid covering sequence for all the graphs in $\mathcal{H}$ is the coordinate-wise maximum of all these sequences.

For large substructures, the sequence $(r_1, r_2, \ldots, r_\tau)$ can be long or include large numbers, and this will affect the computational complexity of RNP-GNNs. For small, e.g., constant-size substructures, the recursion parameters are also small (i.e., $r_i = O(1)$ for all $i$), raising the hope to count these structures efficiently. In particular, $r_1$ is an important parameter. In Section 5.3, we analyze the complexity of RNP-GNNs in more detail.

## 5.2 A Universal Approximation Result for Local Functions

Theorem 1 shows that RNP-GNNs can count substructures if their recursion parameters are chosen carefully. Next, we provide a universal approximation result, which shows that they can represent any function related to local neighborhoods or small subgraphs in a graph.

First, we recall that for a graph $G$, $G(\mathcal{S})$ denotes the subgraph of $G$ induced by the set of nodes $\mathcal{S}$.

**Definition 4.** *A function $\ell : \mathbb{G}_n \to \mathbb{R}^d$ is called an $r-$local graph function if*

$$\ell(G) = \phi(\{\!\!\{ \psi(G(\mathcal{S})) : \mathcal{S} \subseteq \mathcal{V}, |\mathcal{S}| \leq r \}\!\!\}), \tag{11}$$

*where $\psi : \mathbb{G}_r \to \mathbb{R}^{d'}$ is a function on graphs, $\phi$ is a multi-set function, and $\mathcal{V}$ denotes the set of all nodes.*

---

[2]For simplicity, we assume that $\mathcal{H}$ only contains $\tau + 1$ node graphs. If $\mathcal{H}$ includes graphs with strictly less than $\tau + 1$ vertices, we can simply append a sufficient number of zeros to their covering sequences.

[3]One can also generalize this theorem to wider classes of graphs; see Remark 1. However, here in this paper we focus on a special class of graph with covering sequences to keep the results simple and insightful

In other words, a local function only depends on small substructures.

**Theorem 2.** *For any $r-$local graph function $\ell(.)$, there exists an RNP-GNN $f(.;\theta)$ with recursion parameters $(r-1, r-2, \ldots, 1)$ such that $f(G;\theta) = \ell(G)$ for any $G \in \mathbb{G}_n$.*

To prove the theorem, we use specific aggregation functions, but since MLPs with ReLU activation are universal approximators, we can approximate those aggregations and find an RNP-GNN $f(G;\theta)$ implemented by MLPs such that $|f(G;\theta) - \ell(G)| < \epsilon$, for arbitrary small $\epsilon$. As a result, we can provably approximate all the local information in a graph with an appropriate RNP-GNN. Note that we still need recursions, because the function $\psi(.)$ may be an arbitrarily difficult graph function. However, to achieve the full generality of such a universal approximation result, we need to consider large recursion parameters ($r_1 = r - 1$) and injective aggregations in the RNP-GNN network. For universal approximation, we may also need high dimensions if fully connected layers are used for aggregation (see the proof in Appendix B for more details).

As a remark, for $r = n$, achieving universal approximation on graphs implies solving the graph isomorphism problem. But, in this extreme case, the computational complexity of RNP is in general not polynomial in $n$.

### 5.3 COMPUTATIONAL COMPLEXITY

The computational complexity of RNP-GNNs is graph-dependent. For instance, we need to compute the set of local neighborhoods, which is cheaper for sparse graphs. Moreover, in the recursions, we use intersections of neighborhoods which become smaller and sparser.

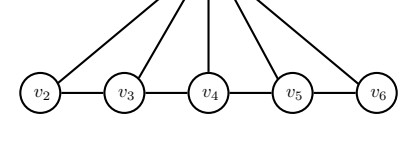

**Theorem 3.** *Let $f(.;\theta) : \mathbb{G}_n \to \mathbb{R}^d$ be an RNP-GNN with recursion parameters $(r_1, r_2, \ldots, r_\tau)$. Assume that the observed graphs $G_1, G_2, \ldots$, whose representations we compute, satisfy the following property: $\max_{v \in [n]} |\mathcal{N}_{r_1}(v)| \leq c$, for a constant $c$. Then the number of node updates in the RNP-GNN is $O(nc^\tau)$.*

Figure 2: For the above graph, $(v_1, v_2, \ldots, v_6)$ is a vertex covering sequence. The corresponding covering sequence $(1, 4, 3, 2, 1)$ is not decreasing.

In other words, if $c = n^{o(1)}$ and $\tau = O(1)$, then RNP-GNN requires relatively few updates (that is, $n^{1+o(1)}$). If the maximum degree of the given graphs is $\Delta$, then $c = O(r_1 \Delta^{r_1})$. Therefore, similarly, if $\Delta = n^{o(1)}$ then we can count with at most $n^{1+o(1)}$ updates. Additional gains may arise from rapidly shrinking neighborhoods, which are not yet accounted for in Theorem 3. To put this in context, the higher-order GNNs based on tensors or $k$-WL would operate on tensors of order $n^{\tau+1}$.

The above results show that when using RNP-GNNs with sparse graphs, we can represent functions of substructures with $k$ nodes without requiring $k-$order tensors. LRPs also encode neighborhoods of distance $r_1$ around nodes. In particular, all $c!$ permutations of the nodes in a neighborhood of size $c$ are considered to obtain the representation.

Table 1: Time complexity of various models. $\Delta$ is the max-degree, and '$-$' means the complexity is not polynomial in $n$.

| Model | worst-case | $\Delta = n^{o(1)}$ | $\Delta = O(\log(n))$ | $\Delta = O(1)$ |
|---|---|---|---|---|
| LRP | $-$ | $-$ | $-$ | $O(n)$ |
| $k-$WL | $n^k$ | $n^k$ | $n^k$ | $n^k$ |
| RNP | $n^k$ | $n^{1+o(1)}$ | $\tilde{O}(n)$ | $O(n)$ |

As a result, LRP networks only have polynomial complexity if $c = o(\log(n))$. Thus, RNP-GNNs can provide an exponential improvement in terms of the tolerable size $c$ of neighborhoods with distance $r_1$ in the graph.

Moreover, Theorem 3 suggests to aim for small $r_1$. The other $r_i$'s may be larger than $r_1$, as shown in Figure 2, but do not affect the upper bound on the complexity.

## 6 AN INFORMATION-THEORETIC LOWER BOUND

In this section, we provide a general information-theoretic lower bound for graph representations that encode a given graph $G$ by first encoding a number of (possibly small) graphs $G_1, G_2, \ldots, G_t$

and then aggregating the resulting representations. The sequence of graphs $G_1, G_2, \ldots, G_t$ may be obtained in an arbitrary way from $G$. For example, in an MPNN, $G_i$ can be the computation tree (rooted tree) at node $i$. As another example, in LRP, $G_i$ is the local neighborhood around node $i$.

Formally, consider a graph representation $f(.; \theta) : \mathbb{G}_n \to \mathbb{R}^d$ as

$$f(G; \theta) = \text{AGGREGATE}(\{\!\{\psi(G_i) : i \in [t]\}\!\}), \quad [t] = \{1, \ldots, t\} \tag{12}$$

for any $G \in \mathbb{G}_n$, where AGGREGATE is a multi-set function, $(G_1, G_2, \ldots, G_t) = \Xi(G)$ where $\Xi(.) : \mathbb{G}_n \to \left(\bigcup_{m=1}^{\infty} \mathbb{G}_m\right)^t$ is a function from one graph to $t$ graphs, and $\psi : \bigcup_{m=1}^{\infty} \mathbb{G}_m \to [s]$ is a function on graphs taking $s$ values. In short, we encode $t$ graphs, and each encoding takes one of $s$ values. We call this graph representation function an $(s, t)$-good graph representation.

**Theorem 4.** *Consider a parametrized class of $(s, t)-$good representations $f(.; \theta) : \mathbb{G}_n \to \mathbb{R}^d$ that is able to count any (not necessarily induced[4]) substructure with $k$ vertices. More precisely, for any graph $H$ with $k$ vertices, there exists $f(.; \theta)$ such that if $C(G_1; H) \neq C(G_2; H)$, then $f(G_1; \theta) \neq f(G_2; \theta)$. Then[5] $t = \tilde{\Omega}(n^{\frac{k}{s-1}})$.*

In particular, for any $(s, t)-$good graph representation with $s = 2$, i.e., binary encoding functions, we need $\tilde{\Omega}(n^k)$ encoded graphs. This implies that, for $s = 2$, enumerating all subgraphs and deciding for each whether it equals $H$ is near optimal. Moreover, if $s \leq k$, then $t = \Theta(n)$ small graphs would not suffice to enable counting.

More interestingly, if $k, s = O(1)$, then it is impossible to perform the substructure counting task with $t = O(\log(n))$. As a result, in this case, considering $n$ encoded graphs (as is done in GNNs or LRP networks) cannot be exponentially improved.

The lower bound in this section is information-theoretic and hence applies to any algorithm. It may be possible to strengthen it by considering computational complexity, too. For binary encodings, i.e., $s = 2$, however, we know that the bound cannot be improved since manual counting of subgraphs matches the lower bound.

## 7 TIME COMPLEXITY LOWER BOUNDS FOR COUNTING SUBGRAPHS

In this section, we put our results in the context of known hardness results for subgraph counting. In general, the subgraph isomorphism problem is known to be NP-complete. Going further, the Exponential Time Hypothesis (ETH) is a conjecture in complexity theory (Impagliazzo & Paturi, 2001), and states that several NP-complete problems cannot be solved in sub-exponential time. ETH, as a stronger version of the $P \neq NP$ problem, is widely believed to hold. Assuming that ETH holds, the $k-$clique detection problem requires at least $n^{\Omega(k)}$ time (Chen et al., 2005). This means that if a graph representation can count *any* subgraph $H$ of size $k$, then computing it requires at least $n^{\Omega(k)}$ time.

**Corollary 1.** *Assuming the ETH conjecture holds, any graph representation that can count any substructure $H$ on $k$ vertices with appropriate parametrization needs $n^{\Omega(k)}$ time to compute.*

The above bound matches the $O(n^k)$ complexity of the higher-order GNNs. Comparing with Theorem 4 above, Corollary 1 is more general, while Theorem 4 has fewer assumptions and offers a refined result for aggregation-based graph representations.

Given that Corollary 1 is a *worst-case* bound, a natural question is whether we can do better for subclasses of graphs. Regarding $H$, even if $H$ is a random Erdös-Rényi graph, it can only be counted in $n^{\Omega(k/\log k)}$ time (Dalirrooyfard et al., 2019).

Regarding the input graph in which we count, consider two classes of sparse graphs: *strongly sparse graphs* have maximum degree $\Delta = O(1)$, and *weakly sparse graphs* have average degree $\bar{\Delta} = O(1)$. We argued in Theorem 3 that RNP-GNNs achieve almost *linear* complexity for the class of strongly sparse graphs. For weakly sparse graphs, in contrast, the complexity of RNP-GNNs is generally not linear, but still polynomial, and can be much better than $O(n^k)$. One may ask whether it is possible

---

[4]The theorem also holds for induced subgraphs, with/without node attributes.

[5]$\tilde{\Omega}(m)$ is $\Omega(m)$ up to poly-logarithmic factors.

to achieve a learnable graph representation such that its complexity for weakly sparse graphs is still linear. Recent results in complexity theory imply that this is impossible:

**Corollary 2** (Gishboliner et al. (2020); Bera et al. (2019; 2020))**.** *There is no graph representation algorithm that runs in linear time on weakly sparse graphs and is able to count any substructure H on k vertices (with appropriate parametrization).*

Hence, RNP-GNNs are close to optimal for several cases of counting substructures with parametrized learnable functions.

## 8 EXPERIMENTS

In this section, we validate our theoretical findings via numerical experiments. Here, we briefly describe our experimental setup and results — further experimental details are given in Appendix H.

Table 2: Numerical results for counting induced triangles and non-induced 3-stars, following the setup of Chen et al. (2020). We report the test MSE divided by variance of the true counts of each substructure (lower is better). The best three models for each task are bolded.

|  | Erdős-Renyi | | Random Regular | |
| --- | --- | --- | --- | --- |
|  | triangle | 3-star | triangle | 3-star |
| GCN | 6.78E-1 | 4.36E-1 | 1.82 | 2.63 |
| GIN | 1.23E-1 | 1.62E-4 | 4.70E-1 | 3.73E-4 |
| GraphSAGE | 1.31E-1 | **2.40E-10** | 3.62E-1 | **8.70E-8** |
| sGNN | 9.25E-2 | 2.36E-3 | 3.92E-1 | 2.37E-2 |
| 2-IGN | 9.83E-2 | 5.40E-4 | 2.62E-1 | 1.19E-2 |
| PPGN | **5.08E-8** | 4.00E-5 | **1.40E-6** | 8.49E-5 |
| LRP-1-3 | 1.56E-4 | 2.17E-5 | 2.47E-4 | **1.88E-6** |
| Deep LRP-1-3 | **2.81E-5** | **1.12E-5** | **1.30E-6** | **2.07E-6** |
| RNP-GNN | **1.39E-5** | **1.39E-5** | **2.38E-6** | 1.50E-4 |

Table 3: Test accuracy on the EXP dataset with setup as in Abboud et al. (2021). Results for baselines taken from Abboud et al. (2021). *Reported PPGN performance differs in other work (Balcilar et al., 2021).

| Model | Accuracy (%) |
| --- | --- |
| GCN-RNI | $98.0 \pm 1.85$ |
| PPGN* | 50.0 |
| 1-2-3-GCN-L | 50.0 |
| 3-GCN | **$99.7 \pm 0.004$** |
| RNP-GNN ($r_1 = 1$) | 50.0 |
| RNP-GNN ($r_1 = 2$) | **$99.8 \pm 0.005$** |

**Counting substructures.** First, we follow the experimental setup of Chen et al. (2020) on tasks for counting substructures. In Table 2, we report results for learning induced subgraph count of triangles and non-induced subgraph count of 3-stars. We test on two datasets of 5000 graphs each: one of Erdős-Renyi graphs and one of random regular graphs. Our RNP-GNN model is consistently within the best performing models for these counting tasks, thus validating our theoretical results. Based on the baseline results taken from (Chen et al., 2020), RNP-GNN tends to widely outperform MPNNs (GCN (Kipf & Welling, 2017), GIN (Xu et al., 2019), GraphSAGE (Hamilton et al., 2017)), and other models not tailored for counting: spectral GNN (Chen et al., 2018), and 2-IGN (Maron et al., 2018). Also, RNP-GNN often beats higher-order GNNs: PPGN (Maron et al., 2019a) and LRP-1-3 (Chen et al., 2020). RNP-GNN is mostly comparable to Deep LRP-1-3, though Deep LRP-1-3 outperforms it in a few cases. Recall that Deep LRP-1-3 is a practical version of LRP — we leave further developments of practical variants of RNP-GNN to future work. The best $r$ parameters for RNP-GNN were: $r = (1, 1, 1, 1)$ for triangles on Erdős-Renyi, $r = (1, 1)$ for stars on Erdős-Renyi, $r = (1, 1)$ for triangles on random regular and $r = (1, 1, 1)$ for stars on random regular.

**Satisfiability of propositional formulas.** Second, we test the expressiveness of our model in distinguishing non-isomorphic graphs that 1-WL cannot distinguish. The EXP dataset of 600 graphs (Abboud et al., 2021) for classifying whether certain propositional formulas are satisfiable requires higher than 1-WL expressive power to achieve better than random accuracy. As shown in Table 3, while our RNP-GNN with $r_1 = 1$ is unable to achieve better than random accuracy, our RNP-GNN with $r_1 = 2$ (we use $r = (2, 1)$ achieves near perfect accuracy — beating all other models based on results taken from (Abboud et al., 2021). These other models include universal models with random node identifiers (GCN-RNI (Abboud et al., 2021)), GNNs with 3-WL power (PPGN (Maron et al., 2019a)), and GNNs that imitate some (possibly weaker) version of 3-WL (1-2-3-GCN-L (Morris et al., 2019), 3-GCN (Abboud et al., 2021)). Thus, our architecture, which is not developed within common frameworks for achieving $k$-WL expressiveness, is in fact powerful at distinguishing non-isomorphic graphs.

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

# A  PROOF OF THEOREM 1

## A.1  PRELIMINARIES

Let us first state a few definitions about the graph functions. Note that for any graph function $f : \mathbb{G}_n \to \mathbb{R}^d$, we have $f(G) = f(H)$ for any $G \cong H$.

**Definition 5.** *Given two graph functions $f, g : \mathbb{G}_n \to \mathbb{R}^d$, we write $f \sqsupseteq g$, if and only if for any $G_1, G_2 \in \mathbb{G}_n$,*

$$\forall G_1, G_2 \in G_n : g(G_1) \neq g(G_2) \implies f(G_1) \neq f(G_2), \tag{13}$$

*or, equivalently,*

$$\forall G_1, G_2 \in G_n : f(G_1) = f(G_2) \implies g(G_1) = g(G_2). \tag{14}$$

**Proposition 1.** *Consider graph functions $f, g, h : \mathbb{G}_n \to \mathbb{R}^d$ such that $f \sqsupseteq g$ and $g \sqsupseteq h$. Then, $f \sqsupseteq h$. In other words, $\sqsupseteq$ is transitive.*

*Proof.* The proposition holds by definition. $\square$

**Proposition 2.** *Consider graph functions $f, g : \mathbb{G}_n \to \mathbb{R}^d$ such that $f \sqsupseteq g$. Then, there is a function $\xi : \mathbb{R}^d \to \mathbb{R}^d$ such that $\xi \circ f = g$.*

*Proof.* Let $\mathbb{G}_n = \sqcup_{i \in \mathbb{N}} \mathcal{F}_i$ be the partitioning induced by the equality relation with respect to the function $f$ on $\mathbb{G}_n$. Similarly define $\mathcal{G}_i, i \in \mathbb{N}$ for $g$. Note that due to the definition, $\{\mathcal{F}_i : i \in \mathbb{N}\}$ is a refinement for $\{\mathcal{G}_i : i \in \mathbb{N}\}$. Define $\xi$ to be the unique mapping from $\{\mathcal{G}_i : i \in \mathbb{N}\}$ to $\{\mathcal{G}_i : i \in \mathbb{N}\}$ which respects the equality relation. One can observe that such $\xi$ satisfies the requirement in the proposition. $\square$

**Definition 6.** *An RNP-GNN is called maximally expressive, if and only if*

- *all the aggregate functions are injective as mappings from a multi-set on a countable ground set to their codomain.*

- *all the combine functions are injective mappings.*

**Proposition 3.** *Consider two RNP-GNNs $f, g$ with the same recursion parameters $\mathbf{r} = (r_1, r_2, \ldots, r_\tau)$ where $f$ is maximally expressive. Then, $f \sqsupseteq g$.*

*Proof.* The proposition holds by definition. $\square$

**Proposition 4.** *Consider a sequence of graph functions $f, g_1, \ldots, g_k$. If $f \sqsupseteq g_i$ for all $i \in [k]$, then*

$$f \sqsupseteq \sum_{i=1}^{k} c_i g_i, \tag{15}$$

*for any $c_i \in \mathbb{R}$, $i \in \mathbb{N}$.*

*Proof.* Since $f \sqsupseteq g_i$, we have

$$\forall G_1, G_2 \in G_n : f(G_1) = f(G_2) \implies g_i(G_1) = g_i(G_2), \tag{16}$$

for all $i \in [k]$. This means that for any $G_1, G_2 \in \mathbb{G}_n$ if $f(G_1) = f(G_2)$ then $g_i(G_1) = g_i(G_2)$, $i \in [k]$, and consequently $\sum_{i=1}^{k} c_i g_i(G_1) = \sum_{i=1}^{k} c_i g_i(G_2)$. Therefore, from the definition we conclude $f \sqsupseteq \sum_{i=1}^{k} c_i g_i$. Note that the same proof also holds in the case of countable summations as long as the summation is bounded. $\square$

**Definition 7.** *Let $H = (\mathcal{V}_H, \mathcal{E}_H, X^H)$ be a attributed connected simple graph with $k$ nodes. For any attributed graph $G = (\mathcal{V}_G, \mathcal{E}_G, X^G) \in \mathbb{G}_n$, the induced subgraph count function $C(G; H)$ is defined as*

$$C(G; H) := \sum_{\mathcal{S} \subseteq [n]} \mathbb{1}\{G(S) \cong H\}. \tag{17}$$

*Also, let $\bar{C}(G; H)$ denote the number of non-induced subgraphs of $G$ which are isomorphic to $H$. It can be defined with the homomorphisms from $H$ to $G$. Formally, if $n > k$ define*

$$\bar{C}(G; H) := \sum_{\substack{\mathcal{S} \subseteq [n] \\ |\mathcal{S}| = k}} \bar{C}(G(\mathcal{S}); H). \tag{18}$$

*Otherwise, $n = k$, and we define*

$$\bar{C}(G; H) := \sum_{\tilde{H} \in \tilde{\mathcal{H}}(H)} c_{\tilde{H}, H} \times \mathbb{1}\{G \cong \tilde{H}\}, \tag{19}$$

*where*

$$\tilde{\mathcal{H}}(H) := \{\tilde{H} \in \mathbb{G}_k : \tilde{H} \sqsupseteq H\}, \tag{20}$$

*is defined with respect to the graph isomorphism, and $c_{\tilde{H}, H} \in \mathbb{N}$ denotes the number of subgraphs in $H$ identical to $\tilde{H}$. Note that $\tilde{\mathcal{H}}(H)$ is a finite set and $\sqsupseteq$ denotes being a (not necessarily induced) subgraph.*

**Proposition 5.** *Let $\mathcal{H}$ be a family of graphs. If for any $H \in \mathcal{H}$, there is an RNP-GNN $f_H(.; \theta)$ with recursion parameters $(r_1, r_2, \ldots, r_\tau)$ such that $f_H \sqsupseteq C(G; H)$, then there exists an RNP-GNN $f(.; \theta)$ with recursion parameters $(r_1, r_2, \ldots, r_\tau)$ such that $f \sqsupseteq \sum_{H \in \mathcal{H}} C(G; H)$.*

*Proof.* Let $f(.; \theta)$ be a maximally expressive RNP-GNN. Note that by the definition $f \sqsupseteq f_H$ for any $H \in \mathcal{H}$. Since $\sqsupseteq$ is transitive, $f \sqsupseteq C(G; H)$ for all $H \in \mathcal{H}$, and using Proposition 4, we conclude that $f \sqsupseteq \sum_{H \in \mathcal{H}} C(G; H)$. $\qquad\square$

The following proposition shows that it is sufficient to address counting attributed graphs.

**Proposition 6.** *Let $H_0$ be an unattributed connected graph. Assume that for any attributed graph $H$, which is constructed by adding arbitrary attributes to $H_0$, there exists an RNP-GNN $f_H(.; \theta_H)$ such that $f_H \sqsupseteq C(G; H)$, then for its unattributed counterpart $H_0$, there exists an RNP-GNN $f(.; \theta)$ with the same recursion parameters as $f_H(.; \theta_H)$ such that $f \sqsupseteq C(G; H_0)$.*

*Proof.* If there exists an RNP-GNN $f_H(.; \theta_H)$ such that $f_H \sqsupseteq C(G; H)$, then for a maximally expressive RNP-GNN $f(.; \theta)$ with the same recursion parameters as $f_H$ we also have $f \sqsupseteq C(G; H)$. Let $\mathcal{H}$ be the set of all attributed graphs $H = (\mathcal{V}, \mathcal{E}, X) \in \mathbb{G}_k$ up to graph isomorphism, where $X \in \mathcal{X}^k$ for a countable set $\mathcal{X}$. Note that $\mathcal{H} = \{H_1, H_2, \ldots\}$ is a countable set. Now we write

$$C(G; H_0) = \sum_{\substack{\mathcal{S} \subseteq [n] \\ |\mathcal{S}| = k}} \mathbb{1}\{G(S) \cong H_0\} \tag{21}$$

$$= \sum_{\substack{\mathcal{S} \subseteq [n] \\ |\mathcal{S}| = k}} \sum_{i \in \mathbb{N}} \mathbb{1}\{G(S) \cong H_i\} \tag{22}$$

$$= \sum_{i \in \mathbb{N}} \sum_{\substack{\mathcal{S} \subseteq [n] \\ |\mathcal{S}| = k}} \mathbb{1}\{G(S) \cong H_i\} \tag{23}$$

$$= \sum_{i \in \mathbb{N}} C(G; H_i). \tag{24}$$

$$\tag{25}$$

Now using Proposition 4 we conclude that $f \sqsupseteq C(G; H_0)$ since $C(G; H_0)$ is always finite. $\qquad\square$

**Definition 8.** *Let $H$ be a (possibly attributed) simple connected graph. For any $\mathcal{S} \subseteq \mathcal{V}_H$ and $v \in \mathcal{V}_H$, define*

$$\bar{d}_H(v; \mathcal{S}) := \max_{u \in \mathcal{S}} d(u, v). \tag{26}$$

**Definition 9.** *Let $H$ be a (possibly attributed) connected simple graph with $k = \tau + 1$ vertices. A permutation of vertices, such as $(v_1, v_2, \ldots, v_{\tau+1})$, is called a vertex covering sequence, with respect to a sequence $\mathbf{r} = (r_1, r_2, \ldots, r_\tau) \in \mathbb{N}^\tau$, called a covering sequence, if and only if*

$$\bar{d}_{H_i'}(v_i; \mathcal{S}_i) \le r_i, \tag{27}$$

*for $i \in [\tau + 1]$, where $H_i' = H(\mathcal{S}_i)$ and $\mathcal{S}_i = \{v_i, v_{i+1}, \ldots, v_{\tau+1}\}$. Let $\mathcal{C}_H(\mathbf{r})$ denote the set of all vertex covering sequences with respect to the covering sequence $\mathbf{r}$ for $H$.*

**Proposition 7.** *For any $G, H \in \mathbb{G}_k$, if $G \sqsupseteq H$ (non-induced subgraph), then*

$$\mathcal{C}_H(\mathbf{r}) \subseteq \mathcal{C}_G(\mathbf{r}), \tag{28}$$

*for any sequence $\mathbf{r}$.*

*Proof.* The proposition follows from the fact that the function $\bar{d}$ is decreasing with introducing new edges. $\qquad\square$

**Proposition 8.** *Assume that Theorem 1 holds for induced-subgraph count functions. Then, it also holds for the non-induced subgraph count functions.*

*Proof.* Assume that for a connected (attributed or unattributed) graph $H$, there exists an RNP-GNN with appropriate recursion parameters $f_H(.; \theta_H)$ such that $f_H \sqsupseteq C(G; H)$, then we prove there exists an RNP-GNN $f(.; \theta)$ with the same recursion parameters as $f_H$ such that $f \sqsupseteq \bar{C}(G; H)$.

If there exists an RNP-GNN $f_H(.; \theta_H)$ such that $f_H \sqsupseteq C(G; H)$, then for a maximally expressive RNP-GNN $f(.; \theta)$ with the same recursion parameters as $f_H$ we also have $f \sqsupseteq C(G; H)$. Note that

$$\bar{C}(G, H) = \sum_{\substack{\mathcal{S} \subseteq [n] \\ |\mathcal{S}| = k}} \bar{C}(G(\mathcal{S}); H) \tag{29}$$

$$= \sum_{\substack{\mathcal{S} \subseteq [n] \\ |\mathcal{S}| = k}} \sum_{\tilde{H} \in \tilde{\mathcal{H}}(H)} c_{\tilde{H}, H} \times \mathbb{1}\{G(S) \cong \tilde{H}\} \tag{30}$$

$$= \sum_{\tilde{H} \in \tilde{\mathcal{H}}(H)} c_{\tilde{H}, H} \sum_{\substack{\mathcal{S} \subseteq [n] \\ |\mathcal{S}| = k}} \mathbb{1}\{G(S) \cong \tilde{H}\} \tag{31}$$

$$= \sum_{i \in \mathbb{N}} c_{H_i, H} \times C(G, H_i), \tag{32}$$

where $\tilde{\mathcal{H}}(H) = \{H_1, H_2, \ldots\}$.

**Claim 1.** $f \sqsupseteq C(G, H_i)$ *for any $i$.*

Using Proposition 4 and Claim 1 we conclude that $f \sqsupseteq \bar{C}(G; H)$ since $\bar{C}(G; H)$ is finite and $f \sqsupseteq C(G, H_i)$ for any $i$, and the proof is complete. The missing part which we must show here is that for any $H_i$ the sequence $(r_1, r_2, \ldots, r_t)$ which covers $H$ also covers $H_i$. This follows from Proposition 7. We are done. $\qquad\square$

At the end of this part, let us introduce an important notation. For any attributed connected simple graph on $k$ vertices $G = (\mathcal{V}, \mathcal{E}, X)$, let $G_v^*$ be the resulting induced graph obtained after removing $v \in \mathcal{V}$ from $G$ with the new attributes defined as

$$X_u^* := (X_u, \mathbb{1}\{(u, v) \in \mathcal{E}\}), \tag{33}$$

for each $u \in \mathcal{V} \setminus \{v\}$. We may also use $X_u^{*v}$ for more clarification.

## A.2 PROOF OF THEOREM 1

We utilize an inductive proof on $\tau$, which is the length of the covering sequence of $H$. Equivalently, due to the definition, $\tau = k - 1$, where $k$ is the number of vertices in $H$. First, we note that due to Proposition 8, without loss of generality, we can assume that $H$ is a simple connected attributed graph and the goal is to achieve the induced-subgraph count function via an RNP-GNN with appropriate recursion parameters. We also consider only maximally expressive networks here to prove the desired result.

**Induction base.** For the induction base, i.e., $\tau = 1$, $H$ is a two-node graph. This means that we only need to count the number of a specific (attributed) edge in the given graph $G$. Note that in this case we apply an RNP-GNN with recursion parameter $r_1 \geq 1$. Denote the two attributes of the vertices in $H$ by $X_1^H, X_2^H \in \mathcal{X}$. The output of an RNP-GNN $f(.;\theta)$ is

$$f(G;\theta) = \phi(\{\!\{\psi(X_v^G, \varphi(\{\!\{X_u^{*v} : u \in \mathcal{N}_{r_1}(v)\}\!\})) : v \in [n]\}\!\}), \tag{34}$$

where we assume that $f(.;\theta)$ is maximally expressive. The goal is to show that $f \sqsupseteq C(G;H)$. Using the transitivity of $\sqsupseteq$, we only need to choose appropriate $\phi, \psi, \varphi$ to achieve $\hat{f} = C(G;H)$ as the final representation. Let

$$\phi(\{\!\{z_v : v \in [n]\}\!\}) := \frac{1}{2 + 2 \times \mathbb{1}\{X_1^H = X_2^H\}} \sum_{i=1}^{n} z_i \tag{35}$$

$$\psi(X, (z, z')) := z \times \mathbb{1}\{X = X_1^H\} + z' \times \mathbb{1}\{X = X_2^H\} \tag{36}$$

$$\varphi(\{\!\{z_u : u \in [n']\}\!\}) := \Big( \sum_{i=1}^{n'} \mathbb{1}\{z_u = (X_2^H, 1)\}, \sum_{i=1}^{n'} \mathbb{1}\{z_u = (X_1^H, 1)\} \Big). \tag{37}$$

Then, a simple computation shows that

$$\hat{f}(G;\theta) = \phi(\{\!\{\psi(X_v^G, \varphi(\{\!\{X_u^{*v} : u \in \mathcal{N}_{r_1}(v)\}\!\})) : v \in [n]\}\!\}), \tag{38}$$

$$= C(G;H). \tag{39}$$

Since $\hat{f}(.;\theta)$ is an RNP-GNN with recursion parameter $r_1$ and for any maximally expressive RNP-GNN $f(.;\theta)$ with the same recursion parameter as $\hat{f}$ we have $f \sqsupseteq \hat{f}$ and $\hat{f} \sqsupseteq C(G;H)$, we conclude that $f \sqsupseteq C(G;H)$ and this completes the proof.

**Induction step.** Assume that the desired result holds for $\tau - 1$ ($\tau \geq 2$). We show that it also holds for $\tau$. Let us first define

$$\mathcal{H}^* := \{H_{v_1}^* : \exists v_2, \ldots, v_\tau \in [k] : (v_1, v_2, \ldots, v_\tau) \in \mathcal{C}_H(\mathbf{r})\} \tag{40}$$

$$c^*(H^0) := \mathbb{1}\{H^0 \in \mathcal{H}^*\} \times \#\{v \in [k] : H_v^* \cong H^0\}, \tag{41}$$

where $H_v^*$ means the induced subgraph after removing a node, with new attributes (see A.1). Note that $\mathcal{H}^* \neq \emptyset$ by the assumption. Let

$$\|\mathcal{H}^*\| := \sum_{H^0 \in \mathcal{H}^*} c^*(H^0). \tag{42}$$

For all $H^0 \in \mathcal{H}^*$, using the induction hypothesis, there is a (universal) RNP-GNN $\hat{f}(.;\hat{\theta})$ with recursion parameters $(r_2, r_3, \ldots, r_\tau)$ such that $\hat{f} \sqsupseteq C(G;H^0)$. Using Proposition 4 we conclude

$$\hat{f} \sqsupseteq \sum_{u \in [k] : H_u^* \in \mathcal{H}^*} C(G;H_u^*). \tag{43}$$

Define a maximally expressive RNP-GNN with the recursion parameters $(r_1, r_2, \ldots, r_\tau)$ as follows:

$$f(G;\theta) = \phi(\{\!\{\psi(X_v^G, \hat{f}(G^*(\mathcal{N}_{r_1}(v)); \hat{\theta})) : v \in [n]\}\!\}). \tag{44}$$

Similar to the proof for $\tau = 1$, here we only need to propose a (not necessarily maximally expressive) RNP-GNN which achieves the function $C(G;H)$.

Let us define

$$f_{H_u^*}(G;\theta) := \phi(\{\!\{\psi_{H_u^*}(X_v^G, \xi \circ \hat{f}(G^*(\mathcal{N}_{r_1}(v)); \hat{\theta})) : v \in [n]\}\!\}), \tag{45}$$

where

$$\phi(\{\!\{z_v : v \in [n]\}\!\}) := \frac{1}{\|\mathcal{H}^*\|} \sum_{i=1}^{n} z_i \tag{46}$$

$$\psi_{H_u^*}(X, z) := z \times \mathbb{1}\{X = X_u^H\}, \tag{47}$$

$$\tag{48}$$

and $\xi \circ \hat{f} = C(G; H_u^*)$. Note that the existence of such function $\xi$ is guaranteed due to Proposition 2. Now we write

$$\|\mathcal{H}^*\| \times C(G; H) = \|\mathcal{H}^*\| \sum_{\mathcal{S} \subseteq [n]} \mathbb{1}\{G(S) \cong H\} \tag{49}$$

$$= \sum_{\mathcal{S} \subseteq [n]} \sum_{v \in \mathcal{S}} \mathbb{1}\{\exists u \in [k] : (G(S \setminus \{v\}))_v^* \cong H_u^* \in \mathcal{H}^* \wedge X_v^G = X_u^H\} \tag{50}$$

$$= \sum_{v \in [n]} \sum_{v \in \mathcal{S} \subseteq [n]} \mathbb{1}\{\exists u \in [k] : (G(S \setminus \{v\}))_v^* \cong H_u^* \in \mathcal{H}^* \wedge X_v^G = X_u^H\} \tag{51}$$

$$= \sum_{v \in [n]} \sum_{v \in \mathcal{S} \subseteq \mathcal{N}_{r_1}(v)} \mathbb{1}\{\exists u \in [k] : (G(S \setminus \{v\}))_v^* \cong H_u^* \in \mathcal{H}^* \wedge X_v^G = X_u^H\}$$

$$\tag{52}$$

$$= \sum_{v \in [n]} \sum_{v \in \mathcal{S} \subseteq \mathcal{N}_{r_1}(v)} \sum_{u \in [k]: H_u^* \in \mathcal{H}^*} \mathbb{1}\{(G(S \setminus \{v\}))_v^* \cong H_u^*\} \mathbb{1}\{X_v^G = X_u^H\} \tag{53}$$

$$= \sum_{v \in [n]} \sum_{u \in [k]: H_u^* \in \mathcal{H}^*} C(G^*(\mathcal{N}_{r_1}(v)); H_u^*) \times \mathbb{1}\{X_v^G = X_u^H\}, \tag{54}$$

which means that

$$\sum_{u \in [k]: H_u^* \in \mathcal{H}^*} f_{H_u^*}(G;\theta) \sqsupseteq C(G; H). \tag{55}$$

However, for a maximally expressive RNP-GNN $f(.;\theta)$ we know that $f \sqsupseteq f_{H_u^*}$ for all $H_u^* \in \mathcal{H}$ and this means that $f \sqsupseteq C(G; H)$. The proof is thus complete.

## B  PROOF OF THEOREM 2

For any attributed graph $H$ on $r$ nodes (not necessarily connected) we claim that RNP-GNNs can count them.

**Claim 2.** *Let $f(.;\theta) : \mathbb{G}_n \to \mathbb{R}^d$ be a maximally expressive RNP-GNN with recursion parameters $(r-1, r-2, \dots, 1)$. Then, $f \sqsupseteq C(G; H)$.*

Now consider the function

$$\ell(G) = \phi(\{\!\{\psi(G(S)) : \mathcal{S} \subseteq \mathcal{V}, |\mathcal{S}| \leq r\}\!\}). \tag{56}$$

We claim that $f \sqsupseteq \ell$ ($f$ is defined in the previous claim) and this completes the proof according to Proposition 2.

To prove the claim, assume that $f(G_1) = f(G_2)$. Then, we conclude that $C(G_1; H) = C(G_2; H)$ for any attributed $H$ (not necessarily connected) with $r$ vertices. Now, we have

$$\ell(G) = \phi(\{\!\{\psi(G(S)) : \mathcal{S} \subseteq \mathcal{V}, |\mathcal{S}| \leq r\}\!\}) \tag{57}$$

$$= \phi(\{\!\{\psi(H) : H \in \mathbb{G}_r, \text{ the multiplicity of } H \text{ is } C(G; H)\}\!\}), \tag{58}$$

which shows that $\ell(G_1) = \ell(G_2)$.

*Proof of Claim 2.* To prove the claim, we use an induction on the number of connected components $c_H$ of graph $H$. If $H$ is connected, i.e., $c_H = 1$, then according to Theorem 1, we know that $f \sqsupseteq C(G; H)$.

Now assume that the claim holds for $c_H = c - 1 \geq 1$. We show that it also holds for $c_H = c$. Let $H_1, H_2, \ldots, H_c$ denote the connected components of $H$. Also assume that $H_i \not\cong H_j$ for all $i \neq j$. We will relax this assumption later. Let us define

$$\mathcal{A}_G := \{(\mathcal{S}_1, \mathcal{S}_2, \ldots, \mathcal{S}_c) : \forall i \in [c] : \mathcal{S}_i \subseteq [n]; G(\mathcal{S}_i) \cong H_i\}. \tag{59}$$

Note that we can write

$$|\mathcal{A}_G| = \prod_{i=1}^{c} C(G; H_i) \tag{60}$$

$$= C(G; H) + \sum_{j=1}^{\infty} c_j' C(G; H_j'), \tag{61}$$

where $H_1', H_2', \ldots$ are all non-isomorphic graphs obtained by adding edges (at least one edge) between $c$ graphs $H_1, H_2, \ldots, H_c$, or contracting a number of vertices of them. The constants $c_j'$ are just used to remove the effect of multiple counting due to the symmetry. Now, since for any $H_i$, $H_j'$ the number of connected components is strictly less that $c$, using the induction, we have $f \sqsupseteq C(G; H_i)$ and $f \sqsupseteq C(G; H_j')$ for all $j$ and all $i \in [c]$. According to Proposition 4, we conclude that $f \sqsupseteq C(G; H)$ and this completes the proof. Also, if $H_i$, $i \in [c]$, are not pairwise non-isomorphic, then we can use $\alpha C(G; H)$ in above equation instead of $C(G; H)$, where $\alpha > 0$ removes the effect of multiple counting by symmetry. The proof is thus complete.

**Remark 1.** *As we explained in the above proof, one can modify Theorem 1 to hold for disconnected graphs. To this end, we need to generalize the notion of covering sequence to hold for this class of graphs. Since this special case is out of scope of this paper, we only restrict to a special, but more insightful case.*

## C    PROOF OF THEOREM 3

To prove Theorem 3, we need to bound the number of node updates required for an RNP-GNN with recursion parameters $(r_1, r_2, \ldots, r_t)$. First of all, we have $n$ variables used for the final representations of vertices. For each vertex $v_1 \in \mathcal{V}$, we explore the local neighborhood $\mathcal{N}_{r_1}(v_1)$ and apply a new RNP-GNN network to that neighborhood. In other words, for the second step we need to update $|\mathcal{N}_{r_1}(v_1)|$ nodes. Similarly, for the $i$th step of the algorithm we have at most

$$\lambda_i := \max_{v_1 \in [n]} \max_{\substack{v_{j+1} \in \mathcal{N}_{r_j}(v_j) \\ \forall j \in [i-1]}} |\mathcal{N}_{r_1}(v_1) \cap \mathcal{N}_{r_2}(v_2) \cap \mathcal{N}_{r_3}(v_3) \ldots \cap \mathcal{N}_{r_i}(v_i)|, \tag{62}$$

updates. Therefore, we can bound the number of node updates as

$$n \times \prod_{i=1}^{\tau} \lambda_i. \tag{63}$$

Since $\lambda_i$ is decreasing in $i$, the desired result holds.

## D    PROOF OF THEOREM 4

Let $K_k$ denote the complete graph on $k$ vertices.

**Claim 3.** *For any $k, n \in \mathbb{N}$, such that $n$ is sufficiently large,*

$$\left| \{C(G; K_k) : G \in \mathbb{G}_n\} \right| \geq \frac{(cn/(k \log(n/k)) - k)^k}{k!} = \tilde{\Omega}(n^k), \tag{64}$$

*where $c$ is a constant which does not depend on $k, n$.*

In particular, we claim that the number of different values that $C(G; K_k)$ can take is $n^k$, up to poly-logarithmic factors.

To prove the theorem, we use the above claim. Consider a class of $(s, t)-$good graph representations $f(.; \theta)$ which can count any substructure on $k$ vertices. As a result, $f \sqsupseteq C(G; K_k)$ for an appropriate parametrization $\theta$. By the definition, $f(.)$ must take at least $\left| \{C(G; K_k) : G \in \mathbb{G}_n\} \right|$ different values, i.e.,

$$\left| \{f(G; \theta) : G \in \mathbb{G}_n\} \right| \geq \left| \{C(G; K_k) : G \in \mathbb{G}_n\} \right|. \tag{65}$$

Also,

$$\left| \{f(G; \theta) : G \in \mathbb{G}_n\} \right| \leq \left| \{\{\!\{\psi(G_i) : i \in [t]\}\!\} : G \in \mathbb{G}_n\} \right|, \tag{66}$$

where $(G_1, G_2, \ldots, G_t) = \Xi(G)$. But, $\psi$ can take only $s$ values. Therefore, we have

$$\left| \{C(G; K_k) : G \in \mathbb{G}_n\} \right| \leq \left| \{f(G; \theta) : G \in \mathbb{G}_n\} \right| \tag{67}$$

$$\leq \left| \{\{\!\{\psi(G_i) : i \in [t]\}\!\} : G \in \mathbb{G}_n\} \right| \tag{68}$$

$$\leq \left| \{\{\!\{\alpha_i : i \in [t]\}\!\} : \forall i \in [t] : \alpha_i \in [s]\} \right| \tag{69}$$

$$\leq (t+1)^{s-1}. \tag{70}$$

As a result, $(t+1)^{s-1} = \tilde{\Omega}(n^k)$ or $t = \tilde{\Omega}(n^{\frac{k}{s-1}})$. To complete the proof, we only need to prove the claim.

*Proof of Claim 3.* Let $p_1, p_2, \ldots, p_m$ be distinct prime numbers less than $n/k$. Using the prime number theorem, we know that $\lim_{n \to \infty} \frac{m}{n/(k \log(n/k))} = 1$. In particular, we can choose $n$ large enough to ensure $cn/(k \log(n/k)) < m$ for any constant $c < 1$.

For any $\mathcal{B} = \{b_1, b_2, \ldots, b_k\} \subseteq [m]$, define $G_{\mathcal{B}}$ as a graph on $n$ vertices such that $\mathcal{V}_{G_{\mathcal{B}}} = V_0 \sqcup (\sqcup_{i \in [k]} \mathcal{V}_i)$, and $|\mathcal{V}_i| = p_{b_i}$. Also,

$$e = (u, v) \in G_{\mathcal{B}} \iff \exists i, j \in [m], i \neq j : u \in \mathcal{V}_i \ \& \ v \in \mathcal{V}_j. \tag{71}$$

The graph $G_{\mathcal{B}}$ is well-defined since $\sum_{i=1}^k p_{b_i} \leq k \times n/k = n$. Note that $C(G_{\mathcal{B}}; K_k) = \prod_{i=1}^k p_{b_i}$. Also, since $p_i, i \in [m]$, are prime numbers, there is a unique bijection

$$\mathcal{B} \overset{\varphi}{\longleftrightarrow} C(G_{\mathcal{B}}; K_k). \tag{72}$$

Therefore,

$$\left| \{C(G; K_k) : G \in \mathbb{G}_n\} \right| \geq \left| \{C(G_{\mathcal{B}}; K_k) : \mathcal{B} \subseteq [m], |\mathcal{B}| = k\} \right| \tag{73}$$

$$= \binom{m}{k} \tag{74}$$

$$\geq \frac{(m-k)^k}{k!} \tag{75}$$

$$\geq \frac{(cn/(k \log(n/k)) - k)^k}{k!}. \tag{76}$$

# E    RELATIONSHIP TO THE RECONSTRUCTION CONJECTURE

Theorem 2 provides a universality result for RNP-GNNs. Here, we note that the proposed method is closely related to the reconstruction conjecture, an old open problem in graph theory. This motivates us to explain their relationship/differences. First, we need a definition for unattributed graphs.

**Definition 10.** *Let $\mathcal{F}_n \subseteq \mathbb{G}_n$ be a set of graphs and let $G_v = G(\mathcal{V} \setminus \{v\})$ for any finite simple graph $G = (\mathcal{V}, \mathcal{E})$, and any $v \in \mathcal{V}$. Then, we say the set $\mathcal{F}$ is reconstructible if and only if there is a bijection*

$$\{\!\{G_v : v \in \mathcal{V}\}\!\} \overset{\Phi}{\longleftrightarrow} G, \tag{77}$$

*for any $G \in \mathcal{F}_n$. In other words, $\mathcal{F}_n$ is reconstructible, if and only if the multi-set $\{\!\{G_v : v \in \mathcal{V}\}\!\}$ fully identifies $G$ for any $G \in \mathcal{F}_n$.*

It is known that the class of disconnected graphs, trees, regular graphs, are reconstructible (Kelly, 1957; McKay, 1997). The general case is still open; however it is widely believed that it is true.

**Conjecture 1** (Kelly (1957)). $\mathbb{G}_n$ *is reconstructible.*

For RNP-GNNs, the reconstruction from the subgraphs $G_v^*$, $v \in [n]$ is possible, since we relabel any subgraph (in the definition of $X^*$) and this preserves the critical information for the recursion to the original graph. In the reconstruction conjecture, this part of information is missing, and this makes the problem difficult. Nonetheless, since in RNP-GNNs we preserve the original node's information in the subgraphs with relabeling, the reconstruction conjecture is not required to hold to show the universality results for RNP-GNNs, although that conjecture is a motivation for this paper. Moreover, if it can be shown that the reconstruction conjecture it true, it may be also possible to find a simple encoding of subgraphs to an original graph and this may lead to more powerful but less complex new GNNs.

## F  THE RNP-GNN ALGORITHM

In this section, we provide pseudocode for RNP-GNNs. The algorithm below computes node representations. In the algorithms, we frequently use MLP modules with ReLU activation. For a graph representation, we can aggregate them with a common readout, e.g., $h_G \leftarrow \text{MLP}\Big(\sum_{v \in \mathcal{V}} h_v^{(k)}\Big)$. Following (Xu et al., 2019), we use sum pooling here, to ensure that we can represent injective aggregation functions.

---

**Algorithm 1** Recursive Neighborhood Pooling-GNN (RNP-GNN)

---

**Input:** $G = (\mathcal{V}, \mathcal{E}, \{x_v\}_{v \in \mathcal{V}})$ where $\mathcal{V} = [n]$, recursion parameters $r_1, r_2, \ldots, r_\tau \in \mathbb{N}$, $\epsilon^{(i)} \in \mathbb{R}$, $i \in [\tau]$, node features $\{x_v\}_{v \in \mathcal{V}}$.
**Output:** $h_v$ for all $v \in \mathcal{V}$
  $h_v^{\text{in}} \leftarrow x_v$ for all $v \in \mathcal{V}$
  **if** $\tau = 1$ **then**

$$h_v \leftarrow \text{MLP}^{(\tau,1)}\Big((1 + \epsilon^{(1)})h_v^{\text{in}} + \sum_{u \in \mathcal{N}_{r_1}(v) \setminus \{v\}} \text{MLP}^{(\tau,2)}(h_u^{\text{in}}, \mathbb{1}(u,v) \in \mathcal{E})\Big),$$

    for all $v \in \mathcal{V}$.
  **else**
    **for** all $v \in V$ **do**
      $G_v' \leftarrow G(\mathcal{N}_{r_1}(v) \setminus \{v\})$, which has node attributes $\{(h_u^{\text{in}}, \mathbb{1}(u,v) \in \mathcal{E})'\}_{u \in \mathcal{N}_{r_1}(v) \setminus \{v\}}$
      $\{\hat{h}_{v,u}\}_{u \in G_v'} \leftarrow \text{RNP-GNN}(G_v', (r_2, r_3, \ldots, r_\tau), (\epsilon^{(2)}, \ldots, \epsilon^{(\tau)}))$
      $h_v \leftarrow \text{MLP}^{(\tau)}\Big((1 + \epsilon^{(\tau)})h_v^{\text{in}} + \sum_{u \in G_v'} \hat{h}_{u,v}\Big)$.
    **end for**
  **end if**
  **return** $\{h_v\}_{v \in \mathcal{V}}$

---

With this algorithm, one can achieve the expressive power of RNP-GNNs if high dimensional MLPs are allowed (Xu et al., 2019; Hornik et al., 1989; Hornik, 1991). That said, in practice, smaller MLPs may be acceptable (Xu et al., 2019).

## G  COMPUTING A COVERING SEQUENCE

As we explained in the context of Theorem 1, we need a covering sequence (or an upper bound to that) to design an RNP-GNN network that can count a given substructure. A covering sequence can be constructed from a spanning tree of the graph.

For reducing complexity, it is desirable to have a covering sequence with minimum $r_1$ (Theorem 3). Here, we suggest an algorithm for obtaining such a covering sequence, shown in Algorithm 2.

For obtaining merely an aribtrary covering sequence, one can compute any minimum spanning tree (MST), and then proceed as with the MST in Algorithm 2.

Given an MST, we build a vertex covering sequence by iteratively removing a leaf $v_i$ from the tree and adding the respective node $v_i$ to the sequence. This ensures that, at any point, the remaining graph is connected. At position $i$ corresponding to $v_i$, the covering sequence contains the maximum distance $r_i$ of $v_i$ to any node in the remaining graph, or an upper bound on that. For efficiency, an upper bound on the distance can be computed in the tree.

To minimize $r_1 = \max_{u \in \mathcal{V}} d(u, v_1)$, we need to ensure that a node in $\arg\min_{v \in \mathcal{V}} \max_{u \in \mathcal{V}} d(u, v)$ is a leaf in the spanning tree. Hence, we first compute $\max_{u \in \mathcal{V}} d(u, v)$ for all nodes $v$, e.g., by running All-Pairs-Shortest-Paths (APSP) (Kleinberg & Tardos, 2006), and sort them in increasing order by this distance. Going down this list, we try whether it is possible to use the respective node as $v_1$, and stop when we find one.

Say $v^*$ is the current node in the list. To compute a spanning tree where $v^*$ is a leaf, we assign a large weight to all the edges adjacent to $v^*$, and a very low weight to all other edges. If there exists such a tree, running an MST with the assigned weights will find one. Then, we use $v^*$ as $v_1$ in the vertex covering sequence. This algorithm runs in polynomial time.

---

**Algorithm 2** Computing a covering sequence with minimum $r_1$

---

**Input:** $H = (\mathcal{V}, \mathcal{E}, X)$ where $\mathcal{V} = [\tau + 1]$
**Output:** A minimal covering sequence $(r_1, r_2 \ldots, r_\tau)$, and its corresponding vertex covering sequence $(v_1, v_2, \ldots, v_{\tau+1})$
  For any $u, v \in \mathcal{V}$, compute $d(u, v)$ using APSP
  $(u_1, u_2, \ldots, u_{\tau+1}) \leftarrow$ all the vertices sorted increasingly in $s(v) := \max_{u \in \mathcal{V}} d(u, v)$
  **for** $i = 1$ to $\tau + 1$ **do**
    Set edge weights $w(u, v) = 1 + \tau \times \mathbb{1}\{u = u_i \vee v = u_i\}$ for all $(u, v) \in \mathcal{E}$
    $H_T \leftarrow$ the MST of $H$ with weights $w$
    **if** $u_i$ is a leaf in $H_T$ **then**
      $v_1 \leftarrow u_i$
      $r_1 \leftarrow s(u_i)$
      **break**
    **end if**
  **end for**
  **for** $i = 2$ to $\tau + 1$ **do**
    $v_i \leftarrow$ one of the leaves of $H_T$
    $r_i \leftarrow \max_{u \in \mathcal{V}_{H_T}} d(u, v_i)$
    $H_T \leftarrow H_T$ after removing $v_i$
  **end for**
  **return** $(r_1, r_2, \ldots, r_\tau)$ and $(v_1, v_2, \ldots, v_{\tau+1})$

---

## H EXPERIMENTAL DETAILS

Table 4: Runtime (in seconds) averaged over 10 epochs of training on the synthetic triangle counting experiments on Erdős-Renyi graphs. Times do not include LRP preprocessing (which takes several minutes for this dataset).

| Model | Parameters | Runtime (s) |
|---|---|---|
| RNP-GNN, $r = (1, 1)$ | 10210 | 153.13 |
| RNP-GNN, $r = (1, 1, 1)$ | 10834 | 199.49 |
| RNP-GNN, $r = (2, 1)$ | 10210 | 370.38 |
| RNP-GNN, $r = (2, 1, 1)$ | 10834 | 835.77 |
| GIN | 10186 | .88 |
| Deep-LRP | 10231 | 24.43 |

## H.1 DATASET AND TASK DETAILS

For the counting experiments, we follow the setup of Chen et al. (2020). There are two datasets: one consisting of 5000 Erdős-Renyi graphs (Erdos et al., 1960) and one consisting of 5000 noisy random regular graphs (Steger & Wormald, 1999). Each Erdős-Renyi graph has 10 nodes, and each random regular graph has either 10, 15, 20, or 30 nodes. Also, $n$ random edges are deleted from each random regular graph, where $n$ is the number of nodes.

For the experiments on distinguishing non-isomorphic graphs, we use the EXP dataset (Abboud et al., 2021). This dataset consists of 600 pairs of graphs (so 1200 graphs in total), where each pair is 1-WL equivalent but distinguishable by 3-WL, and each pair contains one graph that represents a satisfiable formula and one graph that represents an unsatisfiable formula. On average, each graph contains about 44 nodes and 110 edges (Balcilar et al., 2021). We report the mean and standard deviation across 10 cross-validation folds. Additionally, we report the runtimes in Table 4 for the counting experiments.

## H.2 RNP-GNN IMPLEMENTATION DETAILS

Here, we detail some specific design choices we make in implementing our RNP-GNN model. Most embeddings are computed in $\mathbb{R}^d$ for some fixed hidden dimension $d$. The input node features are first embedded in $\mathbb{R}^d$ by an initial linear layer. Then RNP layers are applied to compute node representations. Finally, a sum pooling across nodes followed by a final MLP is used to compute a graph-level output.

An RNP layer for $r = (r_1, \ldots, r_\tau)$ is implemented as follows. Note that the input node features to this layer are in $\mathbb{R}^d$ due to our initial linear layer. Also, note that we concatenate an extra feature dimension due to the augmented indicator feature at each recursion step. To align these feature dimensions, for $l \in [\tau]$, we parameterize the $l$-th GIN (Xu et al., 2019) by a feedforward neural network $\mathrm{MLP}^{(l)} : \mathbb{R}^{d+l} \to \mathbb{R}^{d+l-1}$. For instance, the last GIN has a feedforward network $\mathrm{MLP}^{(\tau)} : \mathbb{R}^{d+\tau} \to \mathbb{R}^{d+\tau-1}$, because after $\tau$ levels of recursion we have augmented $\tau$ features. Dropout and nonlinear activation functions are only applied in the MLPs.

## H.3 HYPERPARAMETERS

For all baseline models, we take the results from other papers. Thus, for the counting experiments the configurations for the baseline models are from Chen et al. (2020), while for the EXP experiments the configurations for the baseline models are from Abboud et al. (2021).

**RNP-GNN hyperparameters.** For all experiments we ran random search over hyperparameters. In all cases we used the Adam optimizer with initial learning rate in $\{.01, .001, .0001, .0005\}$. We train for 100 epochs with a batch size in $\{16, 32, 128\}$. The number of stacked RNP-GNNs for computing node representations is in $\{1, 2\}$. We use a dropout ratio in $\{0, .1, .5\}$. The recursion parameters used varies for each task. We used two layers for each MLP used in the aggregation function. Also, the graph-level output obtained after sum-pooling across nodes is computed by a two layer MLP.

Specifically for the counting experiments, the number of hidden dimensions is searched in $\{16, 32, 64\}$. For all tasks we used $r_1 = 1$. In particular, the optimal $r$ parameters for RNP-GNN were: $r = (1, 1, 1, 1)$ for triangles on Erdős-Renyi, $r = (1, 1)$ for stars on Erdős-Renyi, $r = (1, 1)$ for triangles on random regular and $r = (1, 1, 1)$ for stars on random regular. We use ReLU activations in the MLPs. We either decay the learning rate by half every $25, 50$, or $\infty$ epochs (where $\infty$ means never decaying).

For the EXP experiments, the number of hidden dimensions is searched in $\{8, 16, 32, 64\}$. We use either ELU or ReLU activations in the MLPs. We decay the learning rate by half at the 50th epoch. The recursion parameters are $r = (2, 1)$.

# I    MORE FIGURES

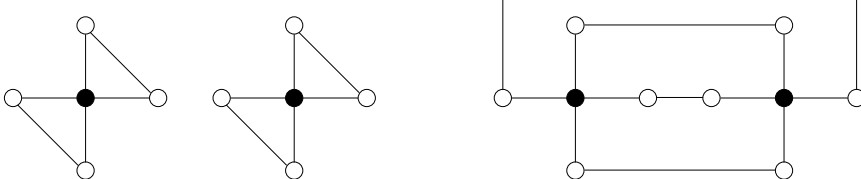

Figure 3: MPNNs cannot count substructures with three nodes or more (Chen et al., 2020). For example, the graph with black center vertex on the left cannot be counted, since the two graphs on the left result in the same node representations as the graph on the right.

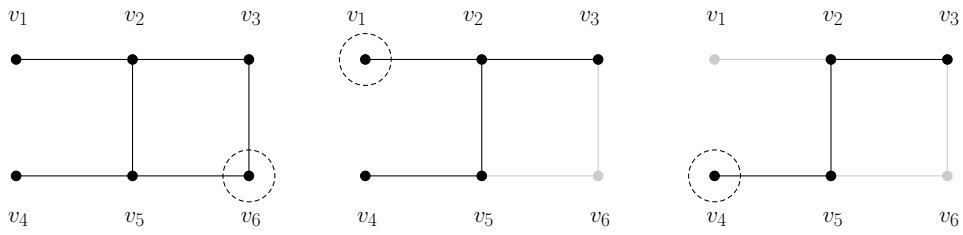

Figure 4: Example of a covering sequence computed for the graph on the left. For this graph, $(v_6, v_1, v_4, v_5, v_3, v_2)$ is a vertex covering sequence with respect to the covering sequence $(3, 3, 3, 2, 1)$. The first two computations to obtain this covering sequence are depicted in the middle and on the right.

