# OpenReview forum: "Counting Substructures with Higher-Order Graph Neural Networks:  Possibility and Impossibility Results"
_ICLR.cc/2022/Conference — ICLR 2022 Submitted_

### Official Review · Reviewer_BQ8d · 2021-10-19

**Correctness:** 3
**Technical Novelty And Significance:** 3
**Empirical Novelty And Significance:** 2
**Recommendation:** 6
**Confidence:** 4

**Main Review:**

I think this paper suggests a very interesting GNN architecture, which essentially allows more expressive power than standard MPNN, while not using k-th order tensors such as in higher-order GNNs. In this aspect, this paper makes a step forward in our understanding of the expressive power of GNNs. Theorem 1 and Theorem 2 successfully provide theoretical guarantees for this architecture, while the experimental part shows that this architecture is indeed successful in tasks in which standard MPNNs fail (such as the EXP dataset, and counting subgraphs). Finally, the information-theoretic and computational complexity lower bounds show that such tasks cannot be solved in a more efficient way. The paper is also well presented and easy to follow.

I have two concerns about the paper which I would be happy to see the author’s response:

1) The comparison of computational complexity to other models in Section 5.3 is unclear, and it is also unclear whether RNP is really more efficient asymptotically than k-WL.
To put into context, suppose \Delta=O(1), and our task at hand is to count a certain subgraph with k nodes. by Table 1, the time complexity of k-WL is n^k, and for RNP it is O(n). But, according to Theorem 1, to count subgraphs with k nodes requires that the \tau in RNP is k-1. This means, according to Theorem 3, that the time complexity (asymptotically) is exponential in k. Hence, it is not clear that RNP is more efficient than k-WL. Also, if k=O(1), then the time complexity of k-WL is n^{O(1)}, which seems to be the same as RNP. I think the authors should give a more concrete case on why RNP is more efficient than higher-order GNNs.

2) The experiments are done only on artificial datasets such as counting subgraphs from random graphs, and the EXP dataset from Abboud et al. which is specifically designed to test 3-WL. It would be beneficial to test whether RNP is successful also on real datasets, otherwise, it is not clear whether RNP can also be used in more realistic settings. For example, testing on datasets from the OGB collection or TUDatasets.

Other minor comments:

1) I think the authors should give more motivation on why counting subgraphs is an interesting task, besides being studied in a previous paper.

2) There are a couple of notations that are used throughout the paper but are not defined, e.g. V(G), \Epsilon(G) G(A) (where A is a subset of nodes), \mathbb{G}_n. They may be standard graph theory notations, but I still suggest adding a short notations section to prevent ambiguity.

3) Although Theorem 2 shows a universal approximation property, the connection to the approximation power of k-WL is unclear. Given a certain choice of (r_1,...,r_\tau), where \tau=k-1, is RNP and k-WL equivalent in approximation power, or is there still a gap? I think this should be clarified in the paper.

4) In Appendix F, r_t-> r_\tau (in the input).

5) In Section 8 it says: “we leave further developments of practical variants of RNP-GNN to future work.” What does it mean? Is RNP as described in the paper not practical?


**Summary Of The Paper:**

The main contribution of this paper is to propose a novel GNN architecture called RNP-GNN. This architecture seems to be specifically designed to tackle the task of counting subgraphs. The authors prove two expressiveness results: (1) RPN-GNN succeeds in the task of counting subgraphs (with an appropriate parameter choice); (2) RPN-GNN has the universal approximation property w.r.t local graph functions. Also, the computational complexity of this architecture is calculated. The authors also provide information-theoretic and computational complexity lower bounds. Finally, several experiments are done on the EXP dataset, and on counting substructures from random graphs.

**Summary Of The Review:**

I think this is a good paper that provides a novel GNN architecture that on the one hand is more expressive than standard MPNN and on the other doesn’t require k-th order tensors to compute. The theoretical aspect of the paper is very thorough and gives insights both into the expressive power of RGP and on lower bounds for the task of counting subgraphs. On the experimental side, the current experiments presented in the paper provide evidence of the expressive power of RGP.
My two concerns are whether RGP is really more computationally efficient than k-WL, and whether it would outperform other MPNNs on standard benchmarks from the GNN literature. I would be happy to read the author’s response and consider raising my score accordingly.

---

> ### Author Response · Authors · 2021-11-19
> **Response to Reviewer BQ8d**
>
>
> We thank the reviewer for acknowledging our “thorough” theoretical contributions: both for our RNP-GNN architecture and for our lower bounds on general representations that count subgraphs. We also appreciate the compliment that our paper is “well presented” and “easy to follow.”
>
> *“The comparison of computational complexity to other models in Section 5.3 is unclear, and it is also unclear whether RNP is really more efficient asymptotically than k-WL.”*
>
> **ANSWER**: In the table, we use asymptotic notation used in Theoretical Computer Science to give a qualitative view of the time complexity in those algorithms. This means ignoring constants, as is standard in theoretical analysis of algorithms. In this analysis, the recursive model is outperforming or matching $k-$WL for asymptotically large $n$.
>
>
> *“​​It would be beneficial to test whether RNP is successful also on real datasets”*
>
> **ANSWER**: See the top-level comment.
>
>
>
> *“I think the authors should give more motivation on why counting subgraphs is an interesting task, besides being studied in a previous paper”*
>
> **ANSWER**: In applications like computational chemistry, materials or drug design, functions we want to learn can depend on substructures (e.g., functional groups). This is mentioned in the paper. Theoretically, the substructure counting is interesting since it characterizes functions that depend merely on local patterns, as opposed to full graph isomorphism.
>
>
>
> *3. “There are a couple of notations that are used throughout the paper but are not defined”*
>
> **ANSWER**: We thank the reviewer for pointing this out, and will clarify these notations.
>
> *“Is RNP as described in the paper not practical?”*
>
> **ANSWER**: See our top-level comment. The framework of recursion currently is not well compatible with typical ways in making deep learning architectures more efficient, e.g., parallelization via matrix operations. Yet, we believe that the practicality can be improved in future work. The focus of this paper is theoretical. It analyzes the addition in representational power gained by recursion; in fact, the use of recursion (more than one level) is new to GNNs, and our results indicate that it can lead to representational benefits. We believe this analysis in itself has merit for the GNN community and its understanding of representational power, and our analysis has the potential to inspire multiple new architectures.

---

> > ### Comment · Reviewer_BQ8d · 2021-11-30
> > **Re: Response**
> >
> > I thank the authors for their response.
> >
> > Regarding the computational complexity, I still think that Table 1 is misleading when comparing the asymptotic behavior of $n$ and $k$.  I understand that the authors only consider the asymptotic of $n$, but just from Table 1 it may seem as if RNP does not have an exponential dependence on $k$, which as I understand is not the case, even when $\Delta = O(1)$. Although this is a somewhat minor issue, as it doesn't affect the results, I think this should be better explained.
> >
> > Regarding the experimental results, I think this is a major issue. I understand that the focus of this paper is on theory, and I acknowledge the theoretical contribution. On the other hand, this paper presents a new GNN architecture that has theoretical guarantees, but without proper experiments, it is difficult to evaluate the benefits of this architecture over existing ones.
> >
> > After reading the rebuttal and the other review, I decided to maintain my score. I still feel like the lack of experiments on real datasets is a major issue, which is also noted in other reviews.
> >
> > There are some relatively small datasets in the TUDataset collection (e.g. Proteins, DD, Enzymes, and many more). I think that experimenting on some of those datasets can be feasible, even without implementing the architecture on GPU. This could give a significantly stronger case on the benefits of the proposed architecture.

---

### Official Review · Reviewer_bVPW · 2021-10-30

**Correctness:** 3
**Technical Novelty And Significance:** 3
**Empirical Novelty And Significance:** 2
**Recommendation:** 5
**Confidence:** 3

**Main Review:**

Overall, this is a practical deep learning contribution, with a specific problem (counting sub-graphs) to solve, despite than a large portion of the paper is spent on theoretical analysis. The novelty mainly lies on how the recursive neighbourhoods are defined. This is not significant and sits in a family of similar structured GNNs. From my view, the paper is borderline on the negative side.

The writing is well polished. However, the notations system is a bit cumbersome and hard to interpret.  The quality of the statements can be improved. For example, it is not clear what is the exact computation of "AGGREGATE" and what is the non-linear activation here.  Theorem 1 is only significant if f outputs an integer, of the difference |f(G_1,\theta)-f(G_2,\theta)| can be lower-bounded.  Similar for theorem 4. Theorem 2 is a weird way to state universal approximation, as we usual states upper-bound on the difference between the neural network and a given function by an arbitrary small \epsilon.

As an important hyper-parameter, the proposed method relies on a vector of radius (r_1, r_2, ... ) that are recursively used in the neighborhood nodes when building the presentation of any node. The overall complexity scales with the size of N_{r_1}, or the size of the
r_1-neighborhoods. In the experiments, $r_1$ is set to a very small value (1 or 2). This is in general an expensive method and only suitable for small-degree graphs, and related application (like counting triangles).

As a practical deep learning contribution, the experiments section is too brief.  First, what is the setting of  (r_1, r_2, ...), which is perhaps the most important parameter, is not specified (instead, there is pointer to Appendix). The statistics of the dataset are also missing, especially, it  is important to say more details about the EXP dataset, and what are "certain propositional formulas" and their sizes.
Second, in the first experiment on counting regular sub-graphs. As Deep-LRP (rather than the authors' method) has the best performance,
it is important to have another experiments to compare their efficiency. Ideally, the authors can show that Deep-LRP is much slower than the proposed RNP-GNN, so the proposed method still wins overall.

#### Minor comments:

The notion of "count substructure" should be clearly defined when it is mentioned in the introduction. What is a substructure, and what is the space when the counting is performed, etc.

Sec 2 the introduction to Higher-Order GNNs can be improved,  as currently it is not so clear what are the underlying operations
of Higher-Order GNNs. I suggest to use some equations similar to eq.(1)

eq.(7) explain what is MLP, and how its parameters/structures are set

eq.(7) explain \epsilon

Section 4, at the end, explain on what settings, the proposed RNP-GNN becomes exactly MPNN.

eq.(8) explain the "\bold{1}" notation, also "\simeq"

Definition 4, explain the difference between {S} and \calligraphic{S}

Definition 4, is \caligraphic{V} introduced in the body of the definition?

**Summary Of The Paper:**

This paper studies the problem of counting isomorphic sub-graphs in a given graph. The author proposed a new graph neural network (GNN), named RNP-GNN, which operates by recursively aggregating node representations based on different settings of the radius. A difference with previous work is that, during recursion, this aggregation is always performed on recursive *subsets* of neighborhoods.  Figure 1 gives an intuitive toy example of the computation process.

The authors also did some theoretical analysis, to show that it is more capable to count sub-graphs, to show that RNP-GNN satisfies universal approximation properties, and to give the exact computation complexity.

The method is tested on counting small structures on random graphs and Erdos-Renyi graphs (which have a very small size, <=30 nodes), and counting non-trivial sub-graphs on the EXP dataset by Abboud et al. (2021), showing better performance than most baselines.

**Summary Of The Review:**

Pro:
- A simple GNN that is well explained
- Good writing

Con:
- Insufficient experimental evaluation
- Quality of mathematical statements

---

> ### Author Response · Authors · 2021-11-20
> **Response to Reviewer bVPW**
>
>
> We thank the reviewer for complimenting our “well polished” writing, and we address their concerns below.
>
> *“Overall, this is a practical deep learning contribution, with a specific problem (counting sub-graphs) to solve, despite than a large portion of the paper is spent on theoretical analysis.”*
>
> AND
>
> *“The novelty mainly lies on how the recursive neighbourhoods are defined. This is not significant and sits in a family of similar structured GNNs. From my view, the paper is borderline on the negative side.”*
>
> **ANSWER**: First, we would like to clarify our contributions.
>
> To your question, beyond the choice of how recursive neighborhoods are defined, the fact that we use recursion at all is very novel. Indeed, basically all current graph neural networks are iterative. Could you please provide examples of “similar structured GNNs” that define neighborhoods in a similar fashion, and/or GNNs that use multiple recursions?
>
> Let us restate our explanation above here. We stress that the main contribution of this paper is to theoretically study “recursion” as a general idea to achieve powerful graph representation algorithms. Recently, this idea has been implicitly/partially used in several papers (we discuss this below) and attracted a lot of attention since it is promising in terms of expressive power. These works, however, take only one recursion step. Instead of focusing on a special recursive algorithm combined with other encodings, we study the category of recursive algorithms and the power gained by recursion alone (with only simple set aggregations), parametrized by recursion parameters. As the main contribution of this paper, we thoroughly analyze those recursive algorithms, and we characterize the expressive power of them in terms of counting substructures. The new architecture (RNPGNN) is near optimal in terms of the complexity of counting for important cases (on sparse graphs).
> To the best of our knowledge, this is the first time that this new hierarchy of recursive algorithms (in the most general case) is introduced and analyzed in the graph representation learning community, and the new theoretical results are insightful for developing better algorithms in the future. The focus of the current work is on theoretical aspects of recursion, and the results suggest a trade-off between computational complexity of the model and the expressive power, which is worth studying independently of practical efficiency. Hence, this work introduces a general, original algorithmic idea for graph representations and characterizes its power. Our results also provide unified insights into the representational power of recursion, even for architectures that may be derived from our ideas in the future. We believe that the originality and theoretical insights by themselves are valuable in providing a better understanding of strategies for graph representations.
>
>
>
> *“The writing is well polished. However, the notations system is a bit cumbersome and hard to interpret. The quality of the statements can be improved.”*
>
> **ANSWER**: Thank you for the compliment on our writing. We are working on making the notation and statements clearer.
>
>
>
> *“For example, it is not clear what is the exact computation of "AGGREGATE" and what is the non-linear activation here.”*
>
> **ANSWER**: We will make it more explicit what AGGREGATE does in our paper. The point is that our framework allows a general aggregation function, just as in general message passing neural networks [Gilmer et al. 2017] [Xu et al. 2019]. For this paper, we work with a most-expressive (i.e. injective) AGGREGATE function, which is the standard one used in GIN [Xu et al. 2019]. The nonlinearities are all in the MLPs, and are usually taken to be ReLU (please see Appendix F and Equation 7 in the paper).
>
>
> [Gilmer et al. 2017] Gilmer J, Schoenholz SS, Riley PF, Vinyals O, Dahl GE. Neural message passing for quantum chemistry. ICML 2017
>
> [Xu et al. 2019] Xu K, Hu W, Leskovec J, Jegelka S. How powerful are graph neural networks?. ICLR 2019
>
>
>
>
> *“Theorem 1 is only significant if f outputs an integer, of the difference |f(G_1,\theta)-f(G_2,\theta)| can be lower-bounded. Similar for theorem 4.”*
>
>
>  **ANSWER**: In our proof of Theorem 1, we give exact formulas for $f(G, \theta)$ such that the statement holds. We are not sure what the reviewer’s exact issue is with this statement.
>
> Theorem 4 merely states that there exists $f$ such that $f(G_1, \theta) \neq f(G_2, \theta)$ if $G_1$ and $G_2$ have different counts for subgraphs of $k$ vertices. Nothing in this statement requires $f$ to output integers or for the differences to be lower-bounded.

---

> > ### Author Response · Authors · 2021-11-20
> > **Response to Reviewer bVPW**
> >
> >
> > *“Theorem 2 is a weird way to state universal approximation, as we usual states upper-bound on the difference between the neural network and a given function by an arbitrary small \epsilon.”*
> >
> > **ANSWER**: We agree that this should be an approximation, and will change the wording accordingly to make it better understandable. The proof goes via refinement, which means discrimination and implies approximation.
> >
> >
> >
> > *“First, what is the setting of (r_1, r_2, ...), which is perhaps the most important parameter, is not specified (instead, there is pointer to Appendix).”*
> >
> > **ANSWER**: We will add to the main paper the exact $r$ we used for the optimal RNPGNN (currently, the $r$ for EXP datasets and the $r_1$ for the counting experiments are listed in Appendix H.3). For the EXP dataset, we use $r=(2,1)$. For the counting experiments we use $r=(1,1,1,1)$ for counting triangles on Erdös-Renyi, $r=(1,1)$ for triangles on random regular, $r=(1,1)$ for stars on Erdös-Renyi, and $r=(1,1,1)$ for stars on random regular. Theorem 1 predicts that $r=(1,1)$ is sufficient to count triangles and $r=(2,1,1)$ is sufficient to count stars, though we find that somewhat adjusting the $r$ can help in learning these particular tasks. Also, $r=(1)$ is in fact sufficient to count stars (see discussion in response to Reviewer T3yH).
> >
> >
> > *“The statistics of the dataset are also missing, especially, it is important to say more details about the EXP dataset, and what are "certain propositional formulas" and their sizes.”*
> >
> > **ANSWER**: The statistics of the datasets are in Appendix H.1. We will add some additional statistics to the main paper and appendix.
> >
> >
> >
> > *“As Deep-LRP (rather than the authors' method) has the best performance, it is important to have another experiments to compare their efficiency.”*
> >
> > **ANSWER**: See our top-level comment. The main goal of this paper is to propose a new hierarchy, and analyze the recursion idea in the most general case with those hyperparameters.
> >
> >
> >
> > *“Minor comments:”*
> >
> > **ANSWER**: We thank the reviewer for these specific notes on our writing and notation. We will make the requisite changes to make our writing more clear.
> >
> > *“​​Definition 4, explain the difference between {S} and \calligraphic{S}”*
> >
> > **ANSWER**: Apologies, there is no difference, this is a typo that we will fix (all of the sets S should be calligraphic).
> >
> > *“Definition 4, is \caligraphic{V} introduced in the body of the definition?”*
> >
> > **ANSWER**: We define it earlier on page 3, though now we remind the reader in Definition 4 what this means.

---

> > > ### Comment · Reviewer_bVPW · 2021-11-30
> > > **Comments after rebuttal**
> > >
> > > I thank the authors for the detailed response.
> > >
> > > I did have acknowledged the novelty as the authors cleverly defined subsets for aggregation.
> > > The authors may agree that even the paper is accepted, the readers may have different
> > > perceptions of the amount of novelty, and this should not be the main point for the rebuttal.
> > > The general idea of "recursion" did already appear in GNNs. For example, a vanilla graph
> > > convolutional network computes node features through recursively aggregating neighbor features.
> > >
> > > Theorem 1: my point is to improve the significance of the statements.
> > > Instead of stating $A\neq{B}$, is that possible to show the gap $|A-B|$ can be lower
> > > bounded? That makes it more non-trivial. This is only a suggestion for potential improvement.
> > >
> > > From the rebuttal, it is still not clear on statistics of the EXP dataset,
> > > which is perhaps the most interesting dataset in the empirical results.
> > >
> > > Comparing with Deep-LRP on efficiency: I still recommend having it to make the experiments more complete.
> > >
> > > The setting of (r_1, r_2, ...): it seems the authors only used the simplest setting (2,1,1,..). More complex settings and sensitivity can be investigated, which is not studied.

---

### Official Review · Reviewer_572g · 2021-11-02

**Correctness:** 3
**Technical Novelty And Significance:** 3
**Empirical Novelty And Significance:** 2
**Recommendation:** 6
**Confidence:** 3

**Main Review:**

Strengths:
* The paper addresses and important problem: most GNN architectures are not expressive enough for many tasks, and architectures that are, are not always feasible. RNP-GNN offers a trade-off between the expressiveness and computational-needs.
* The proposed architecture is backed with wide theoretical analysis, proving that it can count substructures and that it can approximate local graph functions.

Main weaknesses:
* The presentation of the method, described in section 4, was hard to follow, even with the example in Figure 1. I think a clearer explanation would improve the paper.
* According the computational complexity analysis, even with small $c$ and $\tau$ (paragraph after Theorem 3), RNP-GNN will join most higher-order architecture and become unfeasible. I.e., in large, not so sparse graphs the method will not be feasible, yet for small graphs other GNN architectures can be just as good.
* In the experiments sections, I'd like to see a comparison between RNP-GNN and the other architectures with similar performance, so I can see if there is a real improvement w.r.t. accuracy/speed/memory etc.

Addressing the first and last weaknesses will help me change my score.

**Summary Of The Paper:**

While GNN architectures became very popular in recent years, their expressive power is limited. Higher order architectures offer more expressive power, but in turn require sometimes infeasible computation power. This paper offers a new architecture (RNP-GNN) that uses recursive pooling in and offers a trade-off between the expressiveness and the computational needs. The paper provides a theoretical analysis of the proposed architecture, as well as 2 experiments.

**Summary Of The Review:**

To summarize, the paper studies an important problem, and proposes an architecture with strong theoretical justifications. In practice, I'm not convinced that the new architecture has any real advantages over existing, simpler architectures.

I (weakly) support the acceptance of the paper.

---

> ### Author Response · Authors · 2021-11-19
> **Response to Reviewer 572g**
>
> We thank the reviewer for the helpful comments. We have provided a general comment, which partially addresses some of the reviewer’s concerns. In addition, we explain other issues here in the direct reply. We hope that these explanations resolve the reviewer’s comments. The revised version of the paper is improved, thanks to the reviewers’ comments.
>
>
> *“The presentation of the method, described in section 4, was hard to follow, even with the example in Figure 1. I think a clearer explanation would improve the paper.”*
>
> **ANSWER**: Thank you for pointing this out, we will make the presentation of our method clearer in the revision.
>
> *“According the computational complexity analysis, even with small c and $\tau$ (paragraph after Theorem 3), RNP-GNN will join most higher-order architecture and become unfeasible. I.e., in large, not so sparse graphs the method will not be feasible, yet for small graphs other GNN architectures can be just as good.”*
>
> **ANSWER**: As the reviewer correctly mentions, RNPGNNs can join higher-order architectures in terms of complexity. This happens since RNPGNNs are capable of solving the counting problem, and the counting problem is known to get quickly difficult (Section 7). The algorithm is near optimal in the general case. For very special substructures, it may be improvable.
>
> The class of sparse graphs for which RNPGNNs are efficient may appear small. However, in Section 7 we use recent results in complexity theory that show we are at the edge of optimality: even weakly sparse graphs (which are a bit denser than those for which RNPGNN is very efficient) need more than just linear complexity.
>
>
>
> *“In the experiments sections, I'd like to see a comparison between RNP-GNN and the other architectures with similar performance, so I can see if there is a real improvement w.r.t. accuracy/speed/memory etc.”*
>
> **ANSWER**: See our top-level comment. In our current work we focused on theoretical aspects of recursion with respect to representational power, which is a new but important question in GNNs, and has merit in itself. Our experiments are proof-of-concept. Currently, recursion is not well compatible with the common speedups for deep learning, e.g., tensor operations and parallelization and well-engineered toolboxes. But we believe that it is possible to optimize the implementation with further ideas in future work. Our focus is on the theoretical benefit of this algorithmic technique, which can inspire many architectures in future.

---

> > ### Comment · Reviewer_572g · 2021-11-30
> > **After rebuttal**
> >
> > I thank the authors for their response.
> >
> > Although the issues I raised were discussed, I'm still not convinced that the method proposed has any advantage (theoretical or otherwise) over existing NN methods that are in use to solve graph problems.
> >
> > My score remains unchanged.

---

### Official Review · Reviewer_T3yH · 2021-11-02

**Correctness:** 4
**Technical Novelty And Significance:** 4
**Empirical Novelty And Significance:** 2
**Recommendation:** 6
**Confidence:** 5

**Main Review:**

To the best of my knowledge, this is the first time that the algorithmic technique of recursion is employed to define a Graph Neural Network. From a theoretical perspective, the paper is quite strong, and given that substructure counting is arguably a necessary property for a plethora of tasks in graph learning, the universality of RNP in that respect makes the proposed solution quite appealing. Moreover, the computational complexity analysis, as well as the fact that the authors bring impossibility results from the field of complexity theory to the machine learning community, is very insightful.

My major concern right now is the practicality of the method which is reinforced by the fact that the experimental section is quite weak. In particular, in my opinion, the paper raises the reader’s expectations, since the fact that there is a clean procedure to define the recursion neighbourhood sizes creates the impression that it is straightforward to design RNP-GNNs that can perform well not only in counting tasks but also in downstream applications (recently many papers have demonstrated the superiority of GNNs that use information from substructures), albeit with arguably increased computational complexity. However, these expectations are not met, since RNP does not seem to enjoy a clear advantage in simple counting tasks in comparison to other methods, while real-world data are completely omitted.

The above, as well as the fact that RNP is not tested in more challenging tasks, indicate that there might be an inherent practical limitation of RNP, regarding which I am afraid that the authors are not completely upfront. For example, an obvious disadvantage is the exponential dependence in the maximum size of the subgraphs of interest, which might imply that the method is practical only for very small subgraphs. Although the authors argue that this might be hard to improve, there are some cases that this is not true. For example, as far as I know, it is not clear if k-WL algorithms can count *induced* subgraphs of size $>k$, but in some cases, they can achieve this for *not necessarily induced subgraphs*. For example, Arvind et al., FCT’19 and Fürer, CIAC’17 show that 2-WL can count not necessarily induced cycles and paths of up to 7 vertices with $O(n^2)$ complexity, while RNP will need $O(nr_1^6\Delta^{6r_1})$ to solve this task, which will be prohibitive unless the valence of the graph is very small.

At this stage, I am reluctant to suggest a clear acceptance for the paper, since (1) I cannot be sure if it can be implemented in practice and (2) I believe that a stronger experimental section would increase the impact, which will be beneficial both for the authors and the community in general. Below, I provide further details:

**Strengths**:
-	The idea is interesting and novel since it introduces a completely new paradigm for GNNs with clear expressivity advantages.
-	The theoretical results are important, well-posed and in some cases quite general providing new insights into how to prove substructure counting abilities for GNNs. Several constructions and proof techniques are of independent interest which is also positive. For example, the construction of a graph family using prime numbers in Theorem 4, Claim 3, is in my opinion quite clever.
-	The fact that we can define the recursion neighbourhood sizes via the construction of the covering sequences of the subgraphs of interest, provides practical advice and can save the practitioner from tedious hyperparameter tuning.
-	The complexity analysis, although it mainly contains worst-case results, has wide applicability and provides insights into the limitations of GNNs in general.

**Weaknesses**:
- The presentation of RNP does not seem to imply that it is a purely theoretical construction. For example, the argument that the complexity of RNP depends on the sparsity of the graph, contrary to higher-order GNNs is compelling (note that this argument is introduced early on, in the abstract, and this perhaps exceedingly raises expectations). However, the experimental section currently points towards the opposite direction. Can the authors clarify their position w.r.t. that?
- For example, I would expect the triangle counting task to be relatively easy for this network, but RNP does not seem to be better than other more expressive GNNs. The same holds for the 3-star where GraphSage performs much better (question: is GraphSage permutation sensitive here?). How do the authors comment on that? What is the depth of the recursion that you used (is it 2 as expected from the theory)? In general, what is it that prevents RNP from being implemented in a practical form and why isn't it clearly better in these two experiments. Is it only the computational complexity (that perhaps makes it harder to do more extensive experimentation)? Does it have to do with optimization and/or generalisation (it wasn't clear to me if there is a hold-out test here)?
- Unless the method is purely theoretical, I think the authors would benefit a lot from creating a working instantiation of their model and a simple experimental section with more convincing results. Suggestions: (1) counting various substructures and contrasting their r parameters with those predicted by the theory, (2) real-world datasets where substructure counting is important (a single one would be OK – it might be useful here to measure the runtime of your method and contrast it with the sparsity of the graphs)  and optionally (3) predicting graph properties.
- Although the paper is generally well-written, it is notation-heavy and at some points requires significant effort to parse the mathematical expressions. In my opinion, the authors should try to simplify some parts in order to make it more accessible to readers not versed in the theory. For example:
  1.  Architecture - Eq (3-7). This is the heart of the proposed method, but it is a bit hard to follow. Notation can be off-loaded at some points (e.g., Eq.  (5) is very important since it is the recursion formula, yet it’s quite complicated - what is G’? Why do you write h_{u,v}  in Eq. (5) and h_{v,u} in Eq. (6)?)
  2. Could the authors provide an explanation by unrolling the recursion? This might help the reader develop intuition and compare RNP to conventional iterative GNNs.

**Other comments and questions**: (I don’t expect all these to be addressed in the rebuttal, but it might be useful to clarify in the next revision of the paper)
-	How important are the identifiers used to augment the node embeddings (Eq. (3))? Can RNP retain its expressive power without them or it might collapse to that of an MPNN? Regarding the connection with the reconstruction conjecture: How important is the removal of the central node while recursing? In terms of complexity it’s beneficial to remove the central node, but is it crucial in terms of expressive power?
-	What happens in case we want to count a subgraph that is disconnected? How can you define the covering sequence in this case? I think the authors make such an argument in the proof of Theorem 2, but this is not clear from the main paper.
-	I believe at least the proof idea for Theorem 1 should be given in the main paper.
-	As far as I understand, the lower bound of Theorem 4 applies only to functions of the form of Eq. (12), i.e., those that encode $t$ subgraphs with $s$ different unique values and then take the resulting histogram. Is that general enough? Also, although, as I said above, I found the construction of clique-containing graphs very clever, I am wondering if the lower bound is very loose since it is based on counting these graphs only. Finally, personally, I would call it a counting argument rather than information-theoretic (I was expecting to see something else in the proof when I first read it).


**Minor**:
-	You wrote: (Background. Higher-Order GNNs.) “At initialization, each k-tuple is labelled such that two k-tuples are labelled differently if their induced subgraphs are not isomorphic”. If I am not mistaken, k-WL labels k-tuples based on their isomorphism *types*, which is different from isomorphism *classes* since it takes into account the order of the vertices in the tuple as well.
-	The following sentence is unclear: “In particular, we show how the aggregation “augments” local encodings, if they play together and the subgraphs are selected appropriately”.
-	Proposition 6: You state “there is no difference between counting induced attributed graphs and counting induced unattributed graphs in RNP-GNNs”, however, if I am not mistaken your argument does not go both ways. Is that correct?
-	Definition 7 requires further clarifications/is quite hard to follow (e.g., I am not sure if I understand the term homomorphism here since homomorphisms allow vertex repetitions).
-	“However, it is known that MPNNs can count at most star structures or edges”: missing citation: "On Weisfeiler-Leman Invariance: Subgraph Counts and Related Graph Properties", Arvind et al., FCT’19


### --------------- After rebuttal ---------------

My major concerns were not addressed by the rebuttal, hence I will keep my recommendation unchanged. Please see my final comment to the authors for more details.


**Summary Of The Paper:**

This paper proposes a new theoretical paradigm for designing Graph Neural Network architectures, based on recursion. Differently from conventional architectures that iteratively update node embeddings by computing a function of the embeddings of the previous step (locally for MPNNs or globally for higher-order GNNs), *Recursive Neighborhood pooling (RNP)* recursively updates the node embeddings by computing a function of the embeddings of the next level in the recursion hierarchy. Importantly, at each level the computations happen in different neighborhoods in the graph (of potentially different sizes) and node embeddings are enhanced with identifiers that keep track of the graph connectivity.  The authors prove that RNP can count any substructure (induced or not necessarily induced, attributed or not) by appropriately selecting the neighborhood sizes at each recursion level, and show that this implies universal approximation of local functions that depend on substructure counts. Importantly, they provide an explicit procedure to compute the required neighborhood sizes for each substructure of interest, which can potentially be useful for practitioners. In addition, they analyse the computational complexity of RNP and contrast it with theoretical lower bounds, showing that it is close-to-optimal for specific classes of graphs. The proposed framework is tested on two tasks including subgraph counting and graph isomorphism partially verifying the theoretical results.

**Summary Of The Review:**

The paper is quite rich and insightful from a theoretical perspective, but currently, it is unclear if and how it can be implemented in practice and if it can work well. The authors should prepare a much stronger experimental section and clearly discuss the limitations of their approach. I am not negative about the paper, but I believe that this part is crucial to recommend acceptance. I would be willing to discuss my score after that.

---

> ### Author Response · Authors · 2021-11-19
> **Response to Reviewer T3yH**
>
>
> We thank the reviewer for the helpful comments. We have provided a general comment, which partially addresses some of the reviewer’s concerns. In addition, we explain other issues here in the direct reply. We hope that these explanations resolve the reviewer’s comments. The revised version of the paper is improved, thanks to the reviewers’ comments.
>
> *“For example, Arvind et al., FCT’19 and Fürer, CIAC’17 show that 2-WL can count not necessarily induced cycles and paths of up to 7 vertices with $O(n^2)$ complexity, while RNP will need $O(nr_1^6 \Delta^{6 r_1})$ to solve this task, which will be prohibitive unless the valence of the graph is very small.”*
>
> **ANSWER:**  Theorem 1 gives a sufficient condition on the covering sequence for counting subgraphs, but it is not necessary to have recursion parameters $(r_1, …, r_\tau)$ that are a covering sequence for the substructure of interest. For instance, the non-induced count of the star K_{1,m} can be computed by an RNPGNN with recursion parameter (1), where $\tau=1$ and $r_1=1$. This is because this RNPGNN can compute the degree of each node, then the final graph-level readout can compute the star-count from the degree of each node.
>
> Thus, while the condition in Theorem 1 is not necessary, we believe it is an important contribution regardless, as a sufficient condition for counting any (induced or non-induced) substructure, since we focused on the problem of counting substructures in its most general case, and wanted to thoroughly analyze the hierarchy. 2-FWL is sufficient for counting not necessarily induced cycles of up to 7 vertices as shown by Arvind et al. However, there is no nice sufficient condition for counting induced subgraphs, besides trivial and loose sufficient conditions that work at initialization [Chen et al. 2020] (which says that $7$-WL is sufficient to count induced 7-cycles). A sufficient condition for $k$-WL to count non-induced subgraphs is given by the homomorphism-hereditary treewidth [Arvind et al. 2019], but this quantity is significantly more unwieldy than the easily-computable covering number that we introduce.
>
> Nevertheless, the proposed references by the reviewers are still insightful, and we added them to the manuscript as related works.
>
>
>
> *“The presentation of RNP does not seem to imply that it is a purely theoretical construction … Can the authors clarify their position w.r.t. that?”*
>
> **ANSWER**: Please see the top-level comment. In the current version of this paper, we focused on the theoretical understanding of the recursion idea, to see why/how it is even useful, and what is the trade-off between complexity and expressive power. The next step is what is proposed by the review,  and we leave improving the implementation and algorithm, which currently cannot exploit current hardware and deep learning frameworks, as a future work. We think future works can significantly improve upon this. Yet, we believe that the theoretical analysis of recursion as a framework is very valuable in itself, and provides fertile ground for inspiring new architectures.
>
>
>
> *“Question: is GraphSage permutation sensitive here?”*
>
> **ANSWER**: Yes, we take the results from [Chen et al. 2020]. They use an LSTM aggregator in GraphSAGE, which is indeed permutation sensitive (Appendix M.1 of their paper). Of course, this significantly boosts the expressiveness of GraphSAGE, while losing invariance (and thus possibly losing generalization in other tasks).
>
>
>
> *“How important are the identifiers used to augment the node embeddings (Eq. (3))? Can RNP retain its expressive power without them or it might collapse to that of an MPNN?”*
>
> **ANSWER**: For full expressiveness, we indeed need the identifiers, their addition is an important insight. Still, RNPGNN does maintain some expressive power without the augmented identifiers. For instance, consider the graphs $G_1=$ two triangles and $G_2=$ the 6-cycle, which are 1-WL equivalent and thus cannot be distinguished by MPNNs. An RNP-GNN with r_1=1 can distinguish these without the augmented identifiers. This is because on the first recursion level, the 1-neighborhoods of G_1 are two nodes with an edge between them, and the 1-neighborhoods of G_2 consist of two isolated nodes. We will add this illuminating example to the paper.
>
>
>
> *“Regarding the connection with the reconstruction conjecture: How important is the removal of the central node while recursing? In terms of complexity it’s beneficial to remove the central node, but is it crucial in terms of expressive power?”*
>
> **ANSWER**: The reconstruction conjecture also involves removing the “center” node. But, different from the reconstruction conjecture, we retain positional/alignment information by marking the neighbors. The removal and marking gives valuable structural information and, to our knowledge, this combination is not used in other works yet.

---

> > ### Author Response · Authors · 2021-11-19
> > **Response to Reviewer T3yH**
> >
> >
> > *“What happens in case we want to count a subgraph that is disconnected? How can you define the covering sequence in this case? I think the authors make such an argument in the proof of Theorem 2, but this is not clear from the main paper.”*
> >
> > **ANSWER**: The current results are only specific to connected graphs, which we believe are more important to study. However, as the reviewer correctly mentioned, we are not restricted to that class, as we showed in the proof of Theorem 2. The generalized theorem including this case needs “generalized covering sequences” which we believe are out of scope of this paper since it makes the paper more complicated. Nevertheless, we will add explanations to the appendix and cite them in the main body of the paper to make the results more general.
> >
> > *“If I am not mistaken, k-WL labels k-tuples based on their isomorphism types, which is different from isomorphism classes since it takes into account the order of the vertices in the tuple as well.”*
> >
> > **ANSWER**: This is correct, we will fix our wording.
> >
> > *“Proposition 6: You state “there is no difference between counting induced attributed graphs and counting induced unattributed graphs in RNP-GNNs”, however, if I am not mistaken your argument does not go both ways. Is that correct?”*
> >
> > **ANSWER**: We first prove the case of attributed graphs, and then extend to the unattributed case (Proposition 6). Thus, by that sentence we mean “it is sufficient to address counting attributed graphs, because unattributed ones would simply be concluded afterwards”. We apologize for the confusion, and this confusing sentence is now changed in the revision.
> >
> > *“However, it is known that MPNNs can count at most star structures or edges”: missing citation: "On Weisfeiler-Leman Invariance: Subgraph Counts and Related Graph Properties", Arvind et al., FCT’19*
> >
> > **ANSWER**: We added this citation.

---

> > > ### Comment · Reviewer_T3yH · 2021-11-29
> > > **Final comments after rebuttal**
> > >
> > > Although some of the issues I mentioned in my initial review were discussed, I must admit that I found the response insufficient and most of my concerns were not addressed. Below I list the major ones:
> > >  - **Clarity**: Some positive changes have been made, but I think there is still room for improvement in terms of presentation in order to make the paper more accessible to a wider audience (several suggestions have been provided by the reviewers).
> > > - **Experiments**: I have already acknowledged the importance of the theoretical contributions and I do understand that this is the main focus of the paper. However, I am unsatisfied by the discussion w.r.t. the experimental evaluation (the argument that other purely theoretical papers exist is not very convincing, since these analyse existing architectures that *have* been tested in practice, even in limited scenarios, and do not propose a new framework). The authors have put their focus on explaining why their method is currently not amenable to deep learning frameworks and requires further work on the implementation side, but have not discussed the inadequacy of their experiments. My questions in that respect persist:
> > >     - Why didn't RNP excel even in the simple task of triangle counting? Is it because its high complexity prevents more extensive hyperparameter tuning, because of unstable optimisation, or because of generalisation issues? Please clarify in the updated version.
> > >     - Why didn't the authors perform any other experiment, even on a toy dataset? Assuming that the main reason is that the current implementation is highly impractical (judging from the numbers in Table 4), it is, in my opinion, necessary to be clear about this limitation. For example, a separate subsection discussing the limitations and ideas for improvements towards practicality would be helpful. I suggest at least describing the principles of a practical implementation, such as mentioning which parts can be parallelised, and expanding your explanations regarding recent papers that might be considered as instantiations of your framework (e.g., what are the expressivity-practicality tradeoffs in these papers?) - the current discussion is a bit vague.
> > >     - The authors justify the high complexity implied by their bounds by mentioning that this is just a sufficient condition. This is fine, but I am not sure if there is a reason to believe that smaller recursion parameters may work in practice, so this should be tested experimentally.
> > >
> > > Given that my concerns have not been addressed, my views on the paper (and as a result also my score) remain unchanged.

---

### Official Review · Reviewer_vpVz · 2021-11-02

**Correctness:** 3
**Technical Novelty And Significance:** 3
**Empirical Novelty And Significance:** Not applicable
**Recommendation:** 5
**Confidence:** 3

**Main Review:**

The paper presents a fundamental pooling technique that could inspire the construction of future GNN architectures.
The theoretical analysis is quite thorough and places the work in the context of other works.

I have however encountered difficulty reading it and understanding some of it:

1. **Clarity and Readability** - I find the paper packed and hard to follow, although it seems that the authors put a lot of effort into defining every bit of notation.

2. **The algorithm** -
     - It is not clear to me what happens in the recursive pooling calls when the removed vertex disconnects the remaining subgraph?
     - How do the authors choose $(r_1,..._r_{\tau})$ and for instance - how many subgraphs of interest share the same covering sequence? or whether several covering sequences should be used together?

3. **Scalability** - Although providing results on improved complexity, the method is only demonstrated on toy datasets, what are the gaps for scalability in real world datasets?

4. **Larger substructures** - results are shown only on substructures of size 3, I would have wanted to see results on larger ones.

- result in table 3 of PPGN performance seem weird as PPGN expressivity should allow to learn the EXP dataset, see [1]

[1] Balcilar, M. et al. “Breaking the Limits of Message Passing Graph Neural Networks.” ICML (2021).



**Summary Of The Paper:**

Motivated by the limitations of the expressive power of MPNNs this paper addresses one of the expressive shortcomings --- counting substructures.
While previous works allowed counting by suggesting higher-order GNNs, they come at a high computational cost. This paper introduces a flexible recursive pooling technique that provides control over the expressivity-complexity tradeoff. The authors analyze the proposed framework and show expressivity results, complexity results under sparse graph assumptions, and an information-theoretic lower bound. Finally, the paper presents results on toy datasets validating the theoretically guaranteed expressive power.

**Summary Of The Review:**

The paper introduces a novel framework and provides a detailed analysis, but the readability issues are shading it.

---

> ### Author Response · Authors · 2021-11-19
> **Response to reviewer vpVz**
>
>
> We thank the reviewer for the helpful comments. We have provided a general comment, which partially addresses some of the reviewer’s concerns. In addition, we explain other issues here in the direct reply. We hope that these explanations resolve the reviewer’s comments. The revised version of the paper is improved, thanks to the reviewers’ comments.
>
> *1. Clarity and Readability.*
>
> **ANSWER**: In the revision, we are adding further intuition and discussion to the paper to improve both clarity and readability. We will have a particular focus on making the exposition of our algorithm clearer.
>
> *2.1 “Algorithm: what happens in the recursive pooling calls when the removed vertex disconnects the remaining subgraph?”*
>
> **ANSWER**:  When the removed vertex disconnects the remaining subgraph, i.e. $G_v = \mathcal{N}_{r_\tau} \setminus \{v\}$ is disconnected, then when $\tau=1$ all nodes are aggregated over. If $\tau > 1$, then the recursion proceeds just as before: each node is recursively processed, and in effect only pools from its connected component, since it uses (connected) neighborhoods.
>
> *2.2 “Algorithm: How do the authors choose $(r_1,...r_{\tau})$? How many subgraphs of interest share the same covering sequence? Should several covering sequences be used together?”*
>
> **ANSWER**: Many subgraphs of interest share the same covering sequence. For instance, all connected subgraphs on $\tau+1$ nodes admit the covering sequence $(\tau, \tau-1, \ldots, 1)$ (page 5). Indeed, one of the interesting aspects of RNPGNNs is that with a specific covering sequence we can provably count a potentially large set of substructures of interest. In particular, if $H$ admits a covering sequence $r$, and $H_2$ is a graph that contains $H$, i.e.  $H_2$ is formed by adding edges to $H$, then $H_2$ also admits the covering sequence $r$ (end of page 5 to start of page 6).
>
> In terms of complexity, it is ideal to choose the $r$ such that $r_1$ is small, and each other $r_i$ is as small as possible. A simple way to compute an admissible covering sequence (via a spanning tree) is outlined in Appendix G.
>
> We will add to the main paper the exact $r$ we used for the optimal RNPGNNs (currently, the $r$ for EXP datasets and the $r_1$ for the counting experiments are listed in Appendix H.3). For the EXP dataset, we use $r=(2,1)$. For the counting experiments we use $r=(1,1,1,1)$ for counting triangles on Erdos-Renyi, $r=(1,1)$ for triangles on random regular, $r=(1,1)$ for stars on Erdos-Renyi, and $r=(1,1,1)$ for stars on random regular. Theorem 1 predicts that $r=(1,1)$ is sufficient to count triangles and $r=(2,1,1)$ is sufficient to count stars, though we find that somewhat adjusting the $r$ can help in learning these particular tasks. Also, $r=(1)$ is in fact sufficient to count stars (see discussion in response to Reviewer T3yH).
>
> *3.+4. “Scalability: what are the gaps for scalability in real world datasets? Larger substructures: results are shown only on substructures of size 3, I would have wanted to see results on larger ones.”*
>
> **ANSWER**:  See discussion in the top-level comment. Recursion is not yet as amenable to typical speedups of other deep learning frameworks (e.g., parallelization via tensors), however, we believe that accelerations are possible with standard techniques. The experiments currently are proof-of-concept, since the focus of the paper is on theoretical analysis. (Indeed, there exist other purely/mostly theoretical works on GNNs [Garg et al. 2020] [Azizian and Lelarge 2020] [Keriven and Peyre 2019]).
>
>
> [Garg et al. 2020] Garg V, Jegelka S, Jaakkola T. Generalization and representational limits of graph neural networks. ICML 2020
>
> [Azizian and Lelarge 2020] Azizian W, Lelarge M. Expressive power of invariant and equivariant graph neural networks. ICLR 2020
>
> [Keriven and Peyre 2019] Keriven N, Peyré G. Universal invariant and equivariant graph neural networks. NeurIPS 2019
>
>
> *5. “Result in table 3 of PPGN performance seem weird as PPGN expressivity should allow to learn the EXP dataset, see [1]”*
>
> **ANSWER**:  As noted in Section 8 (page 9) of our submission, we took this result directly from Table 1 of the RNI paper [Abboud et al. 2021], which introduced the EXP dataset. We are not sure why there is discrepancy between the empirical results of these two papers, and we will cite [Balcilar et al. 2021] and make note of this in our paper.
>
> [Abboud et al. 2021] Abboud R, Ceylan II, Grohe M, Lukasiewicz T. The surprising power of graph neural networks with random node initialization. IJCAI 2021.

---

> > ### Comment · Reviewer_vpVz · 2021-11-30
> > **After Rebuttal**
> >
> > I thank the authors for their response.
> >
> > Some of the issues I raised were addressed, I do feel however that the paper has to go through major changes in order to make it more accessible to the readers.
> >
> > I keep my score unchanged.

---

### Author Response · Authors · 2021-11-19
**General Comments**

We would like to thank all of the reviewers for their efforts and their helpful comments. Below we make a few general comments about the current results.

**Theory**: We stress that the main contribution of this paper is to theoretically study “recursion” as a general idea to achieve powerful graph representation algorithms. Recently, this idea has been implicitly/partially used in several papers (we discuss this below) and attracted a lot of attention since it is promising in terms of expressive power. Most of these works, however, take only one recursion step. Instead of focusing on a special recursive algorithm combined with other encodings, we study the category of recursive algorithms and the power gained by recursion alone (with only simple set aggregations), parametrized by recursion parameters. As the main contribution of this paper, we thoroughly analyze those recursive algorithms, and we characterize the expressive power of them in terms of counting substructures. The new architecture (RNPGNN) is near optimal in terms of the complexity of counting for important cases (on sparse graphs).
To the best of our knowledge, this is the first time that this new hierarchy of recursive algorithms (in the most general case) is introduced and analyzed in the graph representation learning community, and the new theoretical results are insightful for developing better algorithms in the future. The focus of the current work is on theoretical aspects of recursion, and the results suggest a trade-off between computational complexity of the model and the expressive power, which is worth studying independently of practical efficiency. Hence, this work introduces a general, original algorithmic idea for graph representations and characterizes its power. Our results also provide unified insights into the representational power of recursion, even for architectures that may be derived from our ideas in the future. We believe that the originality and theoretical insights by themselves are valuable in providing a better understanding of strategies for graph representations. Indeed, there exist other purely/mostly theoretical works on GNNs [Garg et al. 2020] [Azizian and Lelarge 2020] [Keriven and Peyre 2019].


**Experiments and implementation**: Several reviewers comment on the experiments section and the practicality of RNPGNN, so we address them simultaneously here. We attach code in the supplemental materials, see rnpgnn.py for the current implementation of our model. That said, the focus of the paper is on theory, and the experiments are proof-of-concept; the implementation is not optimized as we explain below.

Even though the complexity of our architecture is low, our recursive architecture is not yet amenable to computation on current hardware with current deep learning frameworks (we use PyTorch and PyTorch Geometric). Indeed, while other architectures can be expressed in terms of tensor operations (after some preprocessing for LRP), our model must use operations that cannot be expressed as tensor operations, due to the recursions. We include runtimes in Table 4 in our revised paper.

**Practical algorithmic extensions**: Besides small implementation changes, future algorithmic work could well extend RNPGNN to be more practical. Some recent efficient GNNs are similar to RNPGNN with only one recursion layer: GNNs that pass messages on ego-nets (with possibly augmented features) [Bevilacqua et al. 2021] [You et al. 2021] [Zhao et al. 2021], and GNNs based on node reconstruction [Cotta et al. 2021] [Bevilacqua et al. 2021]. These other GNNs modify the graph, then run iterations over the modified graphs, while RNPGNN modifies the graph then recursively modifies the resulting graphs until the last layer of recursion. One might expect benefits from interpolating between these two extremes of iteration and recursion, and expressive gains for these other architectures with more recursions, based on our work. On another note, it may be possible to incorporate stochastic sampling into RNPGNN to more directly speed up the algorithm. All of these interesting questions are left as future work while we focus on theory in this work.

---

> ### Author Response · Authors · 2021-11-19
> **References**
>
>
> [Bevilacqua et al. 2021] Bevilacqua B, Frasca F, Lim D, Srinivasan B, Cai C, Balamurugan G, Bronstein MM, Maron H. Equivariant Subgraph Aggregation Networks. arXiv preprint. 2021.
>
> [You et al. 2021] You J, Gomes-Selman J, Ying R, Leskovec J. Identity-aware graph neural networks. AAAI 2021.
>
> [Zhao et al. 2021] Zhao L, Jin W, Akoglu L, Shah N. From Stars to Subgraphs: Uplifting Any GNN with Local Structure Awareness. arXiv preprint arXiv:2110.03753. 2021 Oct 7.
>
> [Cotta et al. 2021] Cotta L, Morris C, Ribeiro B. Reconstruction for Powerful Graph Representations. NeurIPS 2021.
>
> [Garg et al. 2020] Garg V, Jegelka S, Jaakkola T. Generalization and representational limits of graph neural networks. ICML 2020
>
> [Azizian and Lelarge 2020] Azizian W, Lelarge M. Expressive power of invariant and equivariant graph neural networks. ICLR 2020
>
> [Keriven and Peyre 2019] Keriven N, Peyré G. Universal invariant and equivariant graph neural networks. NeurIPS 2019

---

### Decision · Program_Chairs · 2022-01-20

**Decision:**

Reject

**Comment:**

This submission has been evaluated by 5 reviewers with 3 leaning towards borderline accept and 2 leaning towards borderline reject. Reviewers have been consistently concerned about several aspects of this work, i.e. that *the method is only demonstrated on toy datasets*, that there is an issue with the scalability to larger substructures, that the proposed approach did not excel *in the simple task of triangle counting* or even that *the authors did not perform any other experiments even on a toy dataset*, and that comparisons on Deep-LRP re. efficiency were not provided, and *more complex settings and sensitivity* were not investigated. Reviewers also noted that the general idea of recursion did already appear in GNNs in one or another setting.

In making this decision, AC agrees that there is some potential in the proposed analysis and reviewers also highlighted this as a positive side of the submission. Yet, it is really hard to overlook at the same time the rebuttal where authors had the chance to address all reviewers comments regarding the experiments, their various details, and their variations.

Failing to address these comments to the satisfaction of the majority of reviewers makes it impossible for AC to recommend the acceptance even tough there is every chance that the paper will ultimately make it to a high quality venue after a thorough revision (reviewers have really given a fair number of good suggestions that should assist authors).